# Enhancing extracellular vesicle cargo loading and functional delivery by engineering protein-lipid interactions

Justin A. Peruzzi [1,2,13], Taylor F. Gunnels [2,3,13], Hailey I. Edelstein [1,2], Peilong Lu [4,5,6], David Baker [7,8,9], Joshua N. Leonard [1,3,10,11,12] & Neha P. Kamat [2,3,10,11,12]

Naturally generated lipid nanoparticles termed extracellular vesicles (EVs) hold significant promise as engineerable therapeutic delivery vehicles. However, active loading of protein cargo into EVs in a manner that is useful for delivery remains a challenge. Here, we demonstrate that by rationally designing proteins to traffic to the plasma membrane and associate with lipid rafts, we can enhance loading of protein cargo into EVs for a set of structurally diverse transmembrane and peripheral membrane proteins. We then demonstrate the capacity of select lipid tags to mediate increased EV loading and functional delivery of an engineered transcription factor to modulate gene expression in target cells. We envision that this technology could be leveraged to develop new EV-based therapeutics that deliver a wide array of macromolecular cargo.

The rapid advancement of biological and genetic medicines has spurred research in developing drug delivery systems to transport these molecules to the site of disease[1]. One promising class of vehicles is extracellular vesicles (EVs), which are natural, lipid-based nanoparticles that deliver biologic cargo and mediate natural intercellular communication[1,2]. EVs have low toxicity and immunogenicity profiles[3], which is distinctively useful for scenarios requiring sustained treatment or redosing, making them an attractive platform for delivering medicines in vivo.

Loading therapeutic cargo into EVs while retaining cargo activity and EV integrity remains a challenge. Generally, EVs are loaded with therapeutic cargo using active loading strategies post-harvest[3,4] or by genetically engineering producer cells to produce the molecule of interest in excess[5,6]. Post-harvest strategies such as electroporation of purified EVs enable EVs to be loaded with hydrophobic drugs and nucleic acid cargo. However, such approaches are often challenging to scale up and can introduce purification challenges due to particle aggregation[7]. In general, engineering EV-producing cells to manufacture and directly load the cargo into EVs is simpler and more translatable. A common strategy to drive cargo loading into EVs is to enhance the association of cargo with lipid membranes in the cell. Typically, this is accomplished by fusing protein cargo to transmembrane domains or membrane-targeting sequences, such as lipid anchors, that are known to natively load into EVs[8,9]. Although often effective for loading proteins into EVs, this strategy is not always compatible with the subsequent functional delivery of those proteins

[1]Department of Chemical and Biological Engineering, Northwestern University, Evanston, IL 60208, USA. [2]Center for Synthetic Biology, Northwestern University, Evanston, IL 60208, USA. [3]Department of Biomedical Engineering, Northwestern University, Evanston, IL 60208, USA. [4]Key Laboratory of Structural Biology of Zhejiang Province, School of Life Sciences, Westlake University, Hangzhou, Zhejiang, China. [5]Westlake Laboratory of Life Sciences and Biomedicine, Hangzhou, Zhejiang, China. [6]Institute of Biology, Westlake Institute for Advanced Study, Hangzhou, Zhejiang, China. [7]Department of Biochemistry, University of Washington, Seattle, WA 98195, USA. [8]Institute for Protein Design, University of Washington, Seattle, WA 98195, USA. [9]Howard Hughes Medical Institute, University of Washington, Seattle, WA 98195, USA. [10]Chemistry of Life Processes Institute, Northwestern University, Evanston, IL 60208, USA. [11]Member, Robert H. Lurie Comprehensive Cancer Center, Northwestern University, Evanston, IL 60208, USA. [12]Interdisciplinary Biological Sciences Program, Northwestern University, Evanston, IL 60208, USA. [13]These authors contributed equally: Justin A. Peruzzi, Taylor F. Gunnels. ✉ e-mail: j-leonard@northwestern.edu; nkamat@northwestern.edu

to target cells due to the strong and often irreversible localization of the cargo to an EV membrane. Furthermore, we lack an understanding as to why certain membrane targeting approaches result in improved EV loading while others fail to do so. Exploring and harnessing membrane-protein interactions, which cells utilize to natively traffic and load proteins into EVs, will inform strategies for loading EVs with functional protein cargo.

EVs are enriched with lipids known to form ordered membranes characteristic of lipid rafts−transient ordered lipid domains that play important roles in protein trafficking and signaling[10,11]. While lipid rafts have been hypothesized to play an integral role in EV biogenesis, the link between protein-raft association and protein loading into EVs has not yet been systematically evaluated (Fig. 1a)[10,12–14]. We reasoned that designing proteins which associate with ordered, lipid raft-like membranes should increase loading of EVs with engineered protein cargo.

Here, we explore how protein association with ordered lipid domains can be used to enhance protein loading into EVs. Using membrane spectroscopy and bioinformatics, we confirm that EVs are lipid raft-like in membrane order and protein composition, respectively. We then use bioinformatic analysis to better understand the physical properties characteristic of proteins naturally found in EVs. We use these insights to generate a library of transmembrane and peripheral proteins predicted to load well or poorly into EVs. Using this library, we demonstrate that proteins which associate with the plasma membrane and lipid rafts are effectively loaded into EVs (Fig. 1b). Finally, we show that modulating membrane-protein interactions is a powerful design handle to control EV-loading and functional delivery by using an engineered transcription factor as model cargo. Combined, this work underscores how the ability to understand and modulate membrane biophysical phenomena, such as protein-lipid

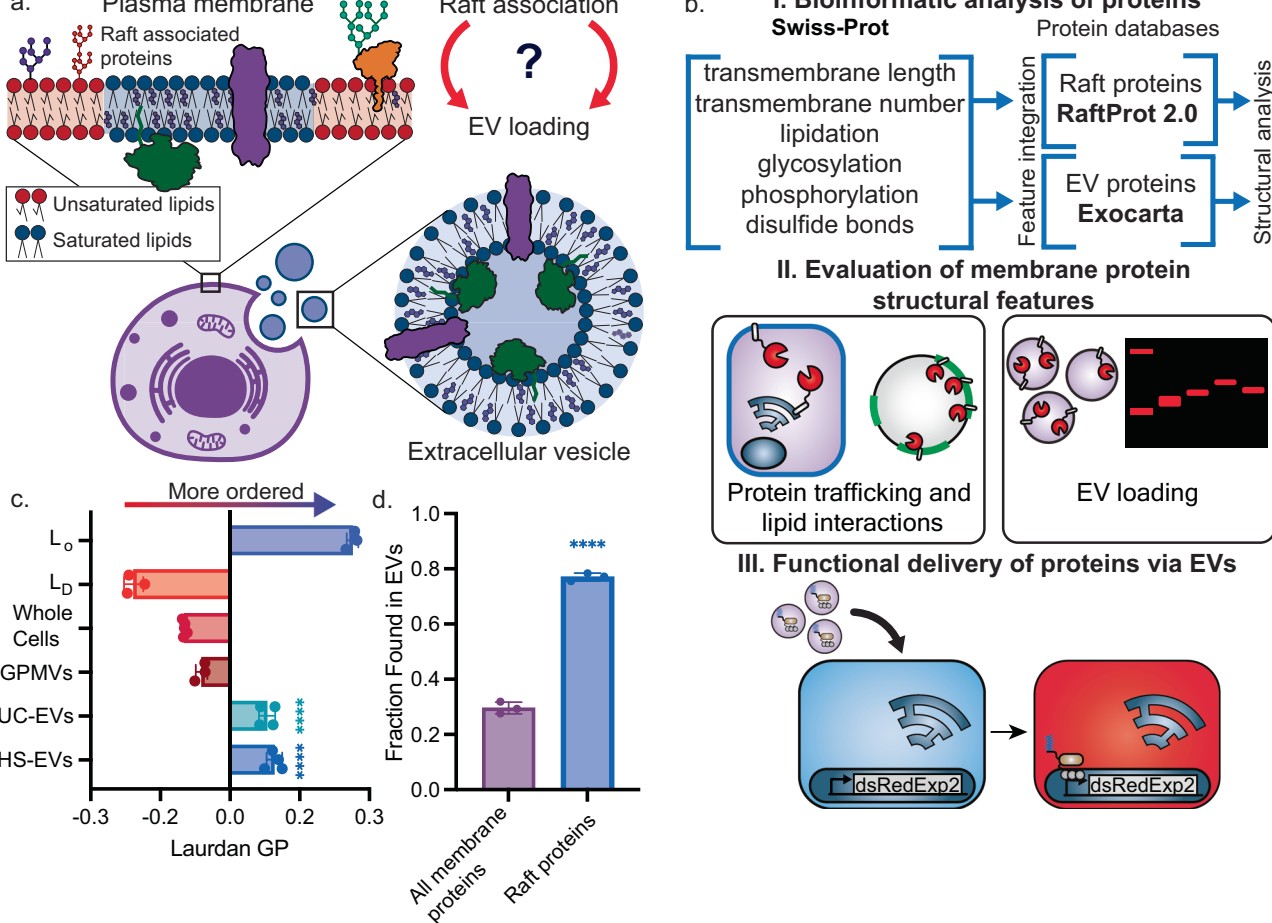

**Fig. 1 | EV membrane physical properties and protein content mirror those of lipid rafts. a** Schematic of our hypothesis that lipid raft association could be used as a handle to load proteins into EVs. **b** Overall analysis and experimental workflow. We used RaftProt 2.0 and Exocarta to understand features of proteins found in EVs and lipid rafts (I); we then built a library of structurally diverse proteins to understand how such features affect protein trafficking, interactions with lipid rafts, and loading into EVs (II); applying these design rules, we demonstrate how lipid-protein interactions can be used to functionally deliver cargo to cells via EVs (III). **c** Laurdan generalized polarization (GP) of ordered liposomes (L_O) composed of 70 mol% DPPC/30 mol% Chol, disordered liposomes (L_D) composed of 70 mol% DOPC/30 mol% Chol, HEK293FT cells, giant plasma membrane vesicles (GPMVs) and vesicles from the high-speed centrifugation EV fraction (HS-EVs) and ultracentrifugation EV fraction (UC-EVs) derived from HEK293FT cells. HS-EV and UC-EVs had high Laurdan GP, similar to L_O membranes as calculated by Eq. 1. Each dot represents an

independent experiment (n ≥ 3). A one-way ANOVA was performed to compare the Laurdan GP of HS-EVs and UC-EVs to the Laurdan GP of all other membranes measured, and comparisons were evaluated using the Sidak multiple comparisons correction. Both EV populations were significantly different from all other vesicle populations except one another (****$p < 0.0001$). **d** The frequency at which human membrane-associated proteins and raft proteins are found in EVs as calculated via bioinformatic analysis. Raft associated proteins are more frequently found in EVs compared to a random selection of human membrane-associated proteins. Each dot represents a separate query of 100 human proteins (n = 3). An unpaired, two-tailed t-test was performed to compare the fraction of all proteins and raft associated proteins found in EVs. The fraction of raft associated proteins found in EVs was significantly different from the fraction of all proteins ($p < 0.0001$). n ≥ 3, error bars represent the standard error of the mean (SEM) throughout the figure. Source data are provided as a Source Data file.

interactions and organization, can be harnessed to engineer functional biologic nanoparticles, which may lead to improved therapeutics.

## Results

### Extracellular vesicles are lipid raft like in membrane physical features and protein composition

To explore the hypothesis that proteins can be loaded into EVs based on protein association with lipid rafts, we first characterized the physical properties of HEK293FT-derived EV membranes. Specifically, we measured the lipid packing of EV membranes via spectroscopy of an environment-sensitive dye, C-laurdan. The spectra of C-laurdan becomes blue-shifted in more ordered, less aqueous environments, which enables membrane order to be quantified[15,16]. Based on previous mass spectrometry characterization of EV membranes, we hypothesized that EVs should be tightly packed, as they are composed largely of lipids which form ordered, raft-like membranes[10,12]. We harvested two subpopulations of EVs from HEK293FTs via differential ultracentrifugation (Supplementary Fig. 1a). Per best practices, these populations are best defined by the separation method used for isolation[17]; we term the population pelleted at 15,000 g as high-speed centrifugation EV fraction (HS-EV) and the population pelleted at 120,416 g as ultracentrifugation EV fraction (UC-EV). To validate our isolation methodology, we have previously confirmed canonical cup-shape EV morphology for both vesicle populations following this purification method via transmission electron microscopy[6]. These populations had the expected 100–200 nm size, were enriched in classic EV markers CD81, CD9, and Alix, and were depleted in the non EV-associated endoplasmic reticulum protein, calnexin (Supplementary Fig. 1b, c). We used C-laurdan to measure membrane order of HS-EVs and UC-EVs and compared it to the membrane order of raft-like, "liquid-ordered" synthetic liposomes ($L_O$, composed of 70 mol% DPPC/30 mol% Chol), non-raft-like "liquid-disordered" liposomes ($L_D$, composed of 70 mol% DOPC/30 mol% Chol), whole HEK293FT cells, and giant plasma membrane vesicles (GPMVs) (Fig. 1c, Supplementary Fig. 2, Supplementary Fig. 3). We found that HS-EVs and UC-EVs were each more ordered than whole cells and GPMVs, and both were similarly ordered to raft-mimetic liposomes, suggesting that both populations of EVs are raft-like. Both vesicle fractions were found to have similar Laurdan GP values, despite the fact that their biogenesis, lipid composition, and protein content may likely differ[12,18–21]. These data corroborate previous analyses of the lipid composition of EVs and demonstrate that EV membranes are highly ordered[10,15].

Since EV membranes are lipid raft-like in lipid order, we next analyzed the proteins reported to associate with EVs using bioinformatics to evaluate whether a relationship exists between protein raft association and EV loading. To guide this effort, we evaluated whether lipid raft associated proteins are more likely to be found in EVs than a group of randomly selected membrane-associated proteins. We leveraged three protein databases for this analysis: Exocarta[22], an EV protein database, RaftProt[23], a raft protein database, and Swiss-Prot, a general annotated database of human proteins. We first calculated the frequency with which 100 randomly selected, human membrane-associated proteins from the Swiss-Prot database appear in EVs. We then repeated this analysis for raft associated proteins (Fig. 1d). Interestingly, we observed that a majority of proteins sampled from the raft protein database were also found in the EV database, and this representation was greater relative to the randomly selected membrane-associated proteins sampled from the Swiss-Prot database. This result suggests that proteins which are known to associate with lipid rafts are more frequently loaded into EV.

### Proteins found in extracellular vesicles exhibit physical features that are known to aid in protein raft association

Given the evidence that EVs are lipid raft-like in lipid order and protein composition, we next sought to identify properties common to proteins found in lipid rafts and extracellular vesicles. We hypothesized that by characterizing the properties of proteins which natively associate with lipid rafts, we could rationally engineer proteins to efficiently load into EVs. We first compiled lists of human proteins reported to associate with lipid rafts and EVs using Raftprot 2.0 and Exocarta, respectively[22,23]. We then integrated protein biophysical and chemical information into our lists of raft and EV proteins using an annotated human protein database (Swiss-Prot). Our new database enabled us to compare protein structural features of proteins found in lipid rafts and EVs to all membrane-associated proteins (Supplementary Table 1). We reasoned that the structural features that are enriched in known raft- or EV-associated proteins, relative to the average for a human protein, likely aid in protein association with raft and EV membranes. Specifically, we looked at the following properties: the presence of a defined transmembrane domain; transmembrane domain length and number; and posttranslational modifications such as lipidation, glycosylation, and phosphorylation[24–29]. These properties were chosen as they affect protein-membrane interactions, cellular trafficking, lipid raft localization, or have been identified as a feature of lipid raft-like proteins[11,24–29].

We performed these analyses for single-pass, multi-pass, and peripheral membrane proteins and found that raft- and EV-associated proteins possessed many similar physical properties and chemical modifications (Fig. 2, Supplementary Figs. 4, 5, 6, 7). To determine which protein features are most likely to enhance protein raft association and EV loading, we confined our search to proteins that localized to the plasma membrane, because the plasma membrane is highly ordered and similar in composition to EV membranes (Fig. 2)[12,15]. We found that raft- and EV-associated single-pass transmembrane proteins had significantly longer average transmembrane domain lengths and higher palmitoylation number compared to the pool of all single-pass transmembrane proteins (Fig. 2a, b). Interestingly, for multi-pass transmembrane proteins, transmembrane domain length and protein palmitoylation were not significantly different between the groups of proteins (Fig. 2c, d). Peripheral membrane proteins that associated with rafts and EVs were further found to possess specific lipidation groups, particularly palmitoyl (raft) and prenyl groups (raft and EV), at a greater than average rate (Fig. 2e–g). To gain further insight, we performed similar analysis on raft proteins found in EVs or not found in EVs (Supplementary Figs. 8, 9, 10, 11). This analysis further emphasized the importance of lipidation, in particular prenylation for peripheral membrane proteins, and native trafficking to the plasma membrane as key factors for loading proteins into EVs. This analysis demonstrates that raft- and EV-associated proteins often possess similar properties and chemical features, bolstering the hypothesis that EVs are enriched in raft-associated proteins (Fig. 1) and providing a blueprint for modifying proteins to enhance loading into EVs.

### Modulating lipid raft affinity of proteins affects cellular trafficking and loading into extracellular vesicles

Once we established features common to proteins found in EVs and lipid rafts, we constructed a library of fluorophore-tagged fusion proteins to experimentally characterize how modulating lipid-protein interactions affects protein loading into EVs. Based on our bioinformatic analysis, we designed transmembrane proteins in which transmembrane domains varied in length, number of passes, geometry, and lipidation state. We similarly designed soluble proteins with different types and amounts of lipidation. Each protein was tagged with a fluorescent moiety—either a monomeric red fluorescent protein (mRFP1[30]) or a HaloTag[31] labeled with a tetramethyl rhodamine (TMR) dye. These fluorescent molecules allowed us to evaluate protein trafficking in live cells, lipid raft association in giant plasma membrane vesicles (GPMVs), and loading into EVs (Fig. 3a).

We began by investigating transmembrane proteins. We first selected the Linker for Activation of T-cells (LAT) as a model

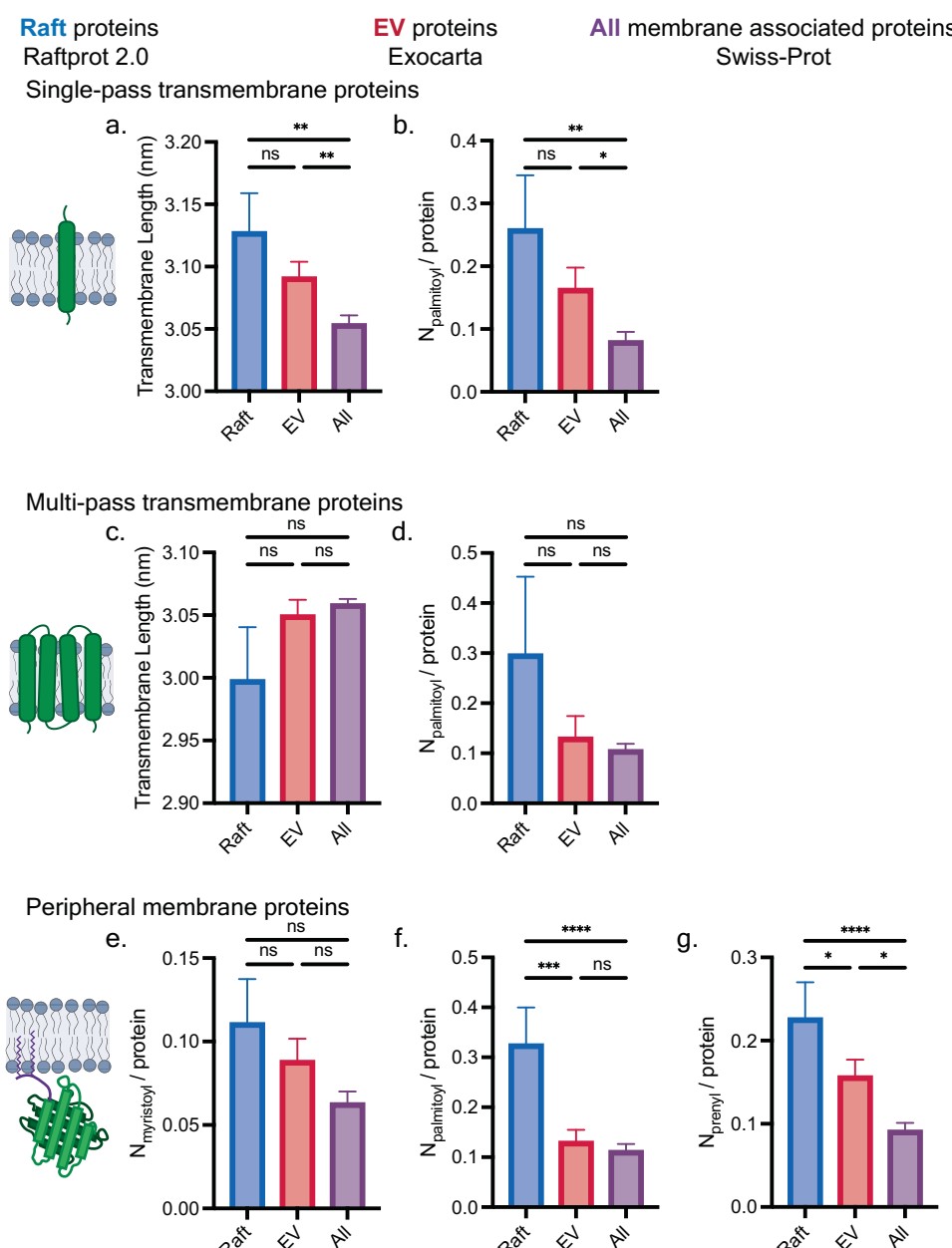

**Fig. 2 | Protein structural features contribute to raft and EV membrane association. a–g** Features of plasma membrane-associated proteins known to associate with lipid rafts and EVs were compared to all human plasma membrane proteins. Average transmembrane domain length and number of palmitoyls per protein ($N_{palmitoyl}$/protein) for (**a**, **b**) single-pass and (**c**, **d**) multi-pass transmembrane proteins, and (**e–g**) number of myristoyl, palmitoyl, and prenyl groups on peripheral membrane proteins were compared. Error bars represent SEM, number of proteins (*n*) in each category can be found in Supplementary Table 1, and this plot showing individual data points is replotted in Supplementary Fig. S4. The Kruskal–Wallis test was performed to compare structural features between each data set, and comparisons were evaluated using the Dunn's multiple comparisons correction (*$p < 0.05$, **$p < 0.01$, ***$p < 0.001$). Source data are provided as a Source Data file.

transmembrane protein to evaluate how protein modifications affect lipid-protein interactions, protein trafficking, and EV loading. LAT naturally associates with lipid rafts and the plasma membrane, making it a useful model protein for this study[25,28]. We generated HaloTag fusion proteins with: wild-type LAT (WT), a LAT with its palmitoylated cysteine mutated to a non-palmitoylated alanine (C26A), LAT with the middle six amino acids removed from its transmembrane domain (dCore), and LAT with amino acid substitutions to increase the transmembrane surface area (High ASA)[25]. We hypothesized that by removing a palmitoylation site, shortening the transmembrane domain, or increasing transmembrane surface area, LAT should (i) decrease its association with the plasma membrane and lipid rafts and (ii) decrease its loading into EVs relative to wild-type LAT (WT)[25].

We evaluated the impact of LAT modifications on protein localization to the plasma membrane and lipid rafts. To these ends, we measured colocalization of protein and membrane-associated dyes in live cells and giant plasma membrane vesicles (GPMVs), respectively. We first transfected HEK293FT cells with each of these constructs and used confocal microscopy with live cells to evaluate the covariance across the image between the protein and either a plasma membrane dye or endoplasmic reticulum dye (calculated using Pearson's coefficients) (Fig. 3b, Supplementary Fig. 12). Consistent with prior reports, we found that wild-type LAT preferentially localized to the plasma membrane more so than the mutants (Fig. 3c, Supplementary Fig. 13a)[28]. To explore how LAT modification impacts association with lipid rafts, we next analyzed LAT localization in GPMVs, vesicles

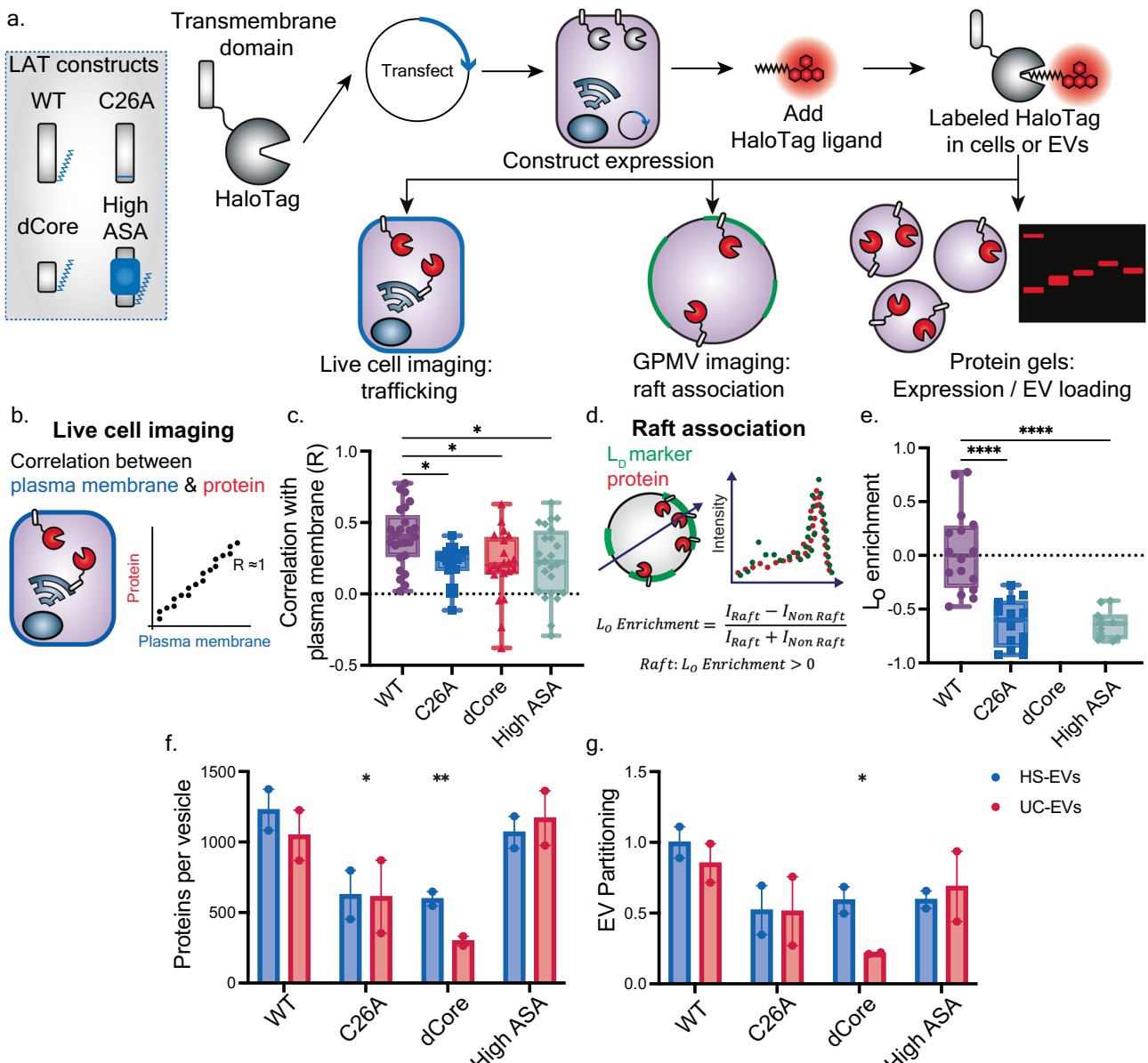

**Fig. 3 | Transmembrane domain lipidation, length, and surface area of LAT affect cellular trafficking, lipid raft association, and extracellular vesicle loading. a** Schematic of workflow to assess cellular trafficking, lipid raft association, and protein loading into EVs for the model transmembrane protein, LAT. For each assay, we compared WT LAT to LAT after the removal of a palmitoyl group (represented in blue) (C26A), LAT with a shortened transmembrane domain (dCore), and LAT with increased hydrophobic surface area (High ASA). **b** Cartoon depicting how confocal images of transfected HEK293FT cells were analyzed. Cells were stained with a plasma membrane dye (Cell Mask) and HaloTag-ligand, protein dye (TMR), imaged, and analyzed pixel-by-pixel to determine colocalization of dyes. **c** Using the approach in (**b**), Pearson's coefficient (R) capturing covariance between the plasma membrane and the HaloTag signal are plotted for each construct. Data are reported in box and whisker plots collected from ≥20 cells from two independent experiments; each symbol is a single cell. The upper and lower bounds represent the minima and maxima of each data measurement, while the box plot marks the lower and upper quartile, as well as the median. A one-way ANOVA was performed to compare plasma membrane localization for each protein, and comparisons were evaluated using Tukey's multiple comparisons correction (*$p < 0.05$). Comparisons not shown were not significant. **d** Cartoon depicting how confocal images of HaloTag-labeled protein in GPMVs were analyzed. Protein was visualized with a HaloTag ligand-conjugated dye (TMR) and the liquid-disordered ($L_D$) region of the GPMV was labeled with DiO. Liquid-ordered ($L_O$) enrichment was calculated

using Eq. 2. **e** Protein partitioning into lipid rafts, here reported as $L_O$ enrichment, for LAT constructs. LAT-dCore was not observed to associate with GPMV membranes, likely due to poor localization to the plasma membrane. Data reported are the average and standard deviation of at least 15 GPMVs from three independent experiments ($n = 15$). The upper and lower bounds represent the minima and maxima of each data measurement, while the box plot marks the lower and upper quartile, as well as the median. Welch's ANOVA test was performed to compare raft localization for each protein, and comparisons were evaluated using the Dunnett T3 multiple comparisons correction (****$p < 0.0001$). Comparisons not shown were not significant. **f** Loading of LAT proteins into EV subpopulations as assessed via SDS-PAGE of fluorescently labeled HaloTag constructs (Supplementary Fig. 15). Each lane of the protein gel received equal numbers of vesicles. $n = 2$ independent sample preparations and protein gels, error bars represent the SEM. **g** EV partitioning of LAT proteins was calculated by normalizing EV protein loading by expression of each construct in EV producer cells as determined by SDS-PAGE (Supplementary Fig. 15). A two-way ANOVA (main effects) was performed to compare proteins per vesicle and EV partitioning, and comparisons were evaluated using Dunnett's multiple comparisons correction. All samples were compared to the respective LAT WT EV subpopulation (*$p < 0.05$). Comparisons not shown were not significant. Values were normalized by WT LAT HS-EV partitioning. $n = 2$ independent sample preparations and protein gels, error bars represent the SEM. Source data are provided as a Source Data file.

derived from cellular plasma membranes that are often used to study protein association with lipid rafts[16,25]. To this end, we transfected HEK293FT cells with each construct and generated GPMVs through the addition of vesiculation agents (paraformaldehyde and dithiothreitol) such that the plasma membrane blebs off from the cell to create protein-incorporated giant vesicles[16]. We visualized protein localization to lipid rafts by labeling GPMVs with DiO—a fluorescent membrane dye which localizes to liquid disordered lipid domains (i.e., non-raft regions)—and cooling vesicles below their miscibility temperature (Fig. 3d, Supplementary Fig. 14). Wild-type LAT partitioned into lipid rafts more so than did the LAT mutants, agreeing with previous studies performed in rat basophilic leukemia (RBL) cells (Fig. 3e)[25]. Surprisingly, a raft-partitioning value could not be measured for LAT dCore because this protein did not localize to the plasma membrane and accordingly did not end up in GPMV membranes. This result agrees with cellular trafficking data in Fig. 3c, but it does not match similar studies performed using RBL cells[25,28]. This discrepancy might suggest that membrane-protein interactions are cell-type specific, perhaps due to differences in membrane composition and membrane physicochemical properties between cell types.

Once we established that removing a palmitoylation site, shortening the transmembrane domain, and increasing transmembrane domain hydrophobic surface area reduced LAT localization to the plasma membrane and association with lipid rafts, we characterized how these changes impacted EV loading. We hypothesized that protein modifications that reduce association with lipid rafts should lead to reduced loading into EVs. We transfected each construct into HEK293FT cells and harvested vesicles from the HS-EV fraction and UC-EV fraction via differential ultracentrifugation. Equal numbers of vesicles were processed via SDS-PAGE, and we quantified protein loading into each vesicle population by probing for the fluorescent TMR-labeled HaloTag on each LAT construct. Notably, wild-type LAT loaded into vesicles from both the HS-EV fraction and UC-EV fraction to a greater extent than did most LAT mutants (Fig. 3f, Supplementary Fig. 15). Protein loading for the C26A and dCore mutants was less than observed for wild-type LAT, which is consistent with the plasma membrane localization and raft partitioning experiments (Fig. 3c, e, Supplementary Fig. 13). Interestingly, the high surface area LAT mutant loaded into EVs nearly as well as did wild-type LAT. However, the high surface area LAT construct also expressed in cells to higher levels than did wild-type LAT (Supplementary Fig. 15). To account for this expression difference, we divided LAT protein loading in EVs by total expression of each respective LAT construct in cells. We then normalized these values by the HS-EV WT LAT value. We term this ratio, 'EV partitioning' (Fig. 3g). Wild-type LAT possessed the highest EV partitioning value. This metric reflects the trends observed for plasma membrane localization and raft partitioning and portrays the effect of lipid-protein interactions on EV loading, independent of effects on overall protein expression/accumulation. Altogether, WT LAT, which most strongly associated with the plasma membrane and lipid rafts, also partitioned most effectively into EVs. Mutations which reduce association with the plasma membrane and lipid rafts, such as the removal of a palmitoylation site or shortening a transmembrane domain, reduced EV loading. These data are reflective of our computational analysis in Fig. 2—longer, palmitoylated single-pass transmembrane domains localize to EV membranes to a greater extent than do shorter, non-palmitoylated proteins—suggesting that such principles may be applied to an array of single-pass transmembrane proteins.

We next investigated whether the trends identified for LAT proteins would hold for different, unrelated transmembrane proteins. To accomplish this, we utilized two sets of de novo designed transmembrane protein complexes—a transmembrane hairpin dimer (4TMD, four transmembrane domains) and a hexameric transmembrane hairpin design (12TMD, twelve transmembrane domains)—which vary

in transmembrane domain length from 24 to 40 Å (Fig. 4a)[32–34]. We reasoned that these proteins are ideal candidates to test the core hypothesis in this study because they are completely synthetic, and therefore we expected that they would be trafficked and loaded into EVs based primarily on physical interactions with membranes, rather than through evolved protein-protein interactions and trafficking mechanisms. Furthermore, evaluating both the 4TMD and 12TMD proteins would enable us to test the extent to which the trends observed for the LAT proteins held for proteins with different numbers of transmembrane domains and different hydrophobic surface area. We fused mRFP1 to the C-terminus of each de novo designed protein and observed how the design choices sampled influenced protein trafficking to the plasma membrane and association with lipid rafts. In live cells, for both protein designs, increasing transmembrane domain length increased plasma membrane localization (Fig. 4b, Supplementary Fig. 13b). This agrees with our characterization of LAT proteins, as well as previous studies demonstrating that protein localization to the plasma membrane is positively correlated with protein transmembrane domain length[28,35]. In GPMV assays, however, none of the constructs appeared to localize to lipid rafts (Fig. 4c). Both our computational analysis in Fig. 2 and previous studies[24,25,29] show that lipidation is a strong determinant of lipid raft association. Thus, it is reasonable that these non-lipidated proteins do not associate with lipid rafts. Like the shortened dCore LAT construct, both 24 Å de novo designed transmembrane domains exhibited reduced localization to GPMV membranes relative to the longer 32 Å, and 40 Å constructs, likely because the former do not traffic well to the plasma membrane.

Once we established how these designed proteins trafficked in cells and evaluated their partitioning into lipid rafts, we characterized loading and partitioning of each set of de novo designed transmembrane proteins into EVs. Transmembrane domain length positively correlated with EV loading and EV partitioning for the proteins with four transmembrane domains (Figs. 4d, e, Supplementary Fig. 16). However, this trend was not observed for the hexameric transmembrane hairpin designs (12TMD). This outcome that transmembrane domain length is not always predictive of EV loading is consistent with our bioinformatic data, since transmembrane domain length did not correlate with raft or EV association for multi-pass transmembrane proteins (Fig. 2c). Furthermore, this trend concords with previous observations by Yurtsever and Lorent[24]. Interestingly, we found that both 40 Å (40 Å 4TMD and 40 Å 12TMD) constructs loaded more effectively into HS-EVs compared to UC-EVs relative to their 24 Å counterparts. This result suggests that lipid-protein interactions may be harnessed to not only increase protein loading into EVs, but that such effects may drive loading into distinct vesicle populations. Combined, the LAT and de novo designed protein data indicate that protein modifications that enhance protein trafficking to the plasma membrane and lipid raft association increase protein loading into EVs.

We next sought to understand how protein engineering to modulate membrane association can be applied to load soluble, cytosolic proteins into EVs. Previous work has demonstrated that lipidating proteins can enhance their loading into EVs[8,9,36]. The types and number of lipids added to proteins can drive localization to different organelle membrane and lipid rafts[27,37–40]. Even with these insights, a link between EV loading, cellular trafficking, and raft association for lipidated, peripheral membrane proteins has not yet been established. Using data compiled in our initial bioinformatic analysis, we selected protein sequences that appeared in either the N- or C-terminal regions of proteins and were natively lipidated. Specifically, we chose protein sequences known to be modified with different amounts and types of lipid moieties (e.g., myristoyl, palmitoyl, farnesyl, geranylgeranyl) and genetically fused these sequences to HaloTag. Hereafter, we refer to these protein sequences and their expected modifications as 'lipid tags.' We generated 5 HaloTag-containing proteins: unlipidated, cytosolic/soluble HaloTag (Sol); a HaloTag with a single N-terminal

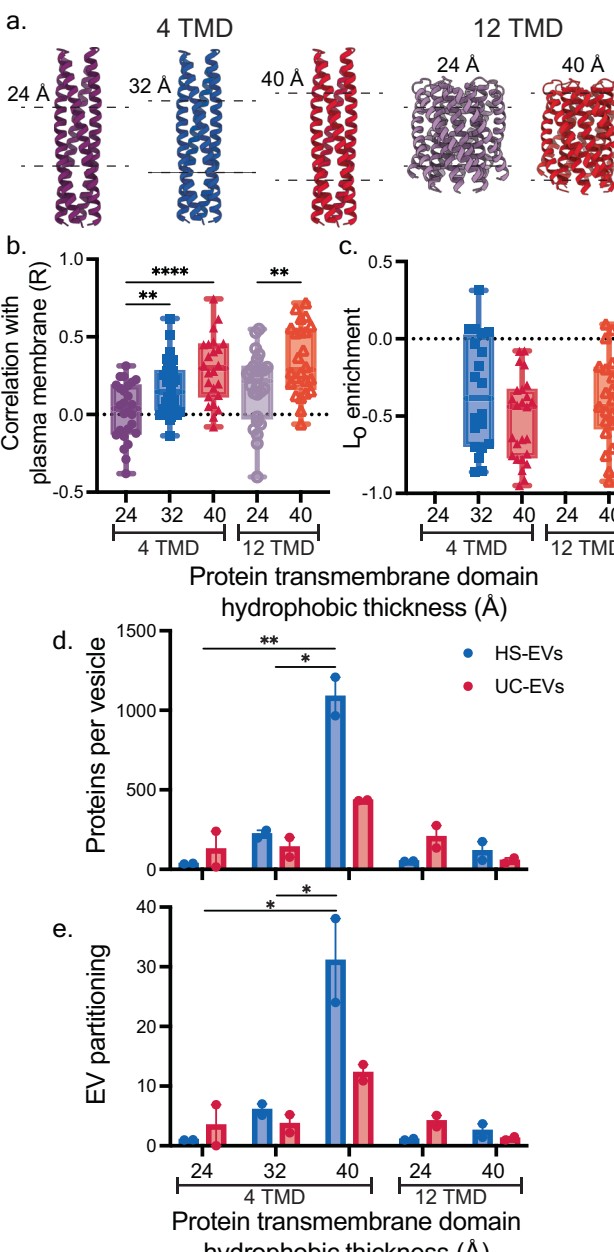

**Fig. 4 | Transmembrane domain length of de novo designed proteins affects cellular trafficking and extracellular vesicle loading. a** Structure of synthetic, de novo designed transmembrane domain (TMD) proteins assessed. Two families of proteins were assessed, either containing four or twelve transmembrane domains. Different lengths of transmembrane domains in each family were assessed. **b** Pearson's coefficient (*R*) from confocal microscopy images evaluating the subcellular localization of mRFP1-tagged de novo designed proteins with a plasma membrane-staining dye, Cell Mask. Data are reported in box and whisker plots collected from 30 cells from two independent experiments; each symbol is a single cell. To compare plasma membrane localization, a one-way ANOVA with Tukey's multiple comparisons was performed on the dimeric 4TMD designs and an unpaired, two-tailed *T*-test was performed on the hexameric 12TMD designs (**$p < 0.01$; ****$p < 0.0001$). **c** Partitioning of de novo designed proteins with liquid-ordered lipid raft regions ($L_O$) of GPMVs as determined by confocal microscopy analysis. Raft association did not significantly differ among de novo designed proteins (one-way ANOVA, Tukey's multiple comparisons correction). Both 24 Å constructs did not associate with GPMV membranes. Data reported are the average and standard deviation of at least 20 GPMVs from three independent experiments (*n* = 20). For (**b**, **c**), the upper and lower bounds represent the minima and maxima of each data measurement, while the box plot marks the lower and upper quartile, as well as the median. **d, e** Protein loading and partitioning into EVs as calculated by western blots probing the 3x FLAG tag (Supplementary Fig. 16). Each lane of the protein gel received equal numbers of vesicles. **e** EV partitioning was calculated by normalizing EV protein loading by expression of each construct in EV producer cells as determined by western blot. EV partitioning values were normalized by 24 Å 4TMD HS-EV partitioning. A two-way ANOVA (main effects) was performed to compare proteins per vesicle and EV partitioning within each protein family (dimeric 4TMD and hexameric 12TMD designs), and comparisons were evaluated using Tukey's multiple comparisons correction (*$p < 0.05$; ****$p < 0.0001$). Samples were compared to the respective 24 Å long transmembrane domain for their protein family. Comparisons not shown were not significant. *n* = *2* independent sample preparations and western blots for (**d**, **e**). Error bars represent the SEM. Source data are provided as a Source Data file.

Once we established how each peripheral membrane protein construct interacted with cellular membranes, we characterized how each protein loaded and partitioned into EVs. We again transfected HEK293FT cells with each construct, harvested EVs, characterized protein loading and partitioning into EVs via SDS-PAGE. Lipidation increased HaloTag loading relative to soluble HaloTag for all conditions in both EV populations, achieving up to 400 proteins per vesicle and a loading improvement of up to ~4x over passive loading (Fig. 5d, Supplementary Fig. 17). Although the G tag did not drive association with lipid rafts, it did confer high EV loading. This result suggests that although lipid raft association can drive protein loading into EVs, it is not required. When normalizing to overall protein expression in the producer cells, we found that the motifs which strongly associated with lipid rafts (i.e., PM, PPF) generally drove the greatest loading of cargo into EVs, achieving a 5–10 X increase in EV partitioning relative to soluble HaloTag (Fig. 5e). Interestingly, we observed that when cells expressed protein constructs that strongly partition to the plasma membrane and into EVs, as many as 50% more vesicles were generated in the HS-EV fraction, but this increase was not observed in the UC-EV fraction; this differential effect correlates with the degree of EV partitioning (Fig. 5f–h). Microvesicles, or EVs which are derived from the plasma membrane, are likely enriched within the HS-EV fraction[18,21]. The observed increase in HS-EVs could be explained by enhanced microvesicle formation due to increased membrane budding as a result of protein crowding on the plasma membrane, and this observation suggests a possible strategy to intentionally increase EV production[41]. Taken together, our data suggest a positive correlation between plasma membrane and lipid raft association and EV loading for cytosolic protein cargo, further validating bioinformatic analysis presented in Fig. 2.

Combined, the data collected in Figs. 3–5 provide insight into how lipid-protein interactions affect protein trafficking and ultimately

myristoylation site (M); a HaloTag with an N-terminal palmitoylation and myristoylation site (PM); a HaloTag with a single C-terminal geranylgeranylation site (G); and a HaloTag with two palmitoylation sites and a single farnesylation site on the C-terminus (PPF) (Fig. 5a). Lipid tags were placed on termini in a manner that matches the native location of the tags (i.e., within the source protein on which the tag is modeled). We transfected plasmids encoding each construct into HEK293FT cells and observed how each protein localized to the endoplasmic reticulum and plasma membrane (Fig. 5b, Supplementary Fig. 13c). As expected, soluble, non-lipidated HaloTag did not localize to the plasma membrane. Furthermore, it appeared that a single lipidation site was sufficient to increase association of cargo protein with the plasma membrane (M, G); the addition of multiple lipid sites strengthened the association with the plasma membrane (PM, PPF). We then investigated how each of these constructs associated with lipid rafts in GPMVs (Fig. 5c). The PM and PPF tags conferred more association with lipid rafts than did the M and G tags. In contrast, soluble HaloTag was observed inside GPMVs but did not associate with GPMV membranes.

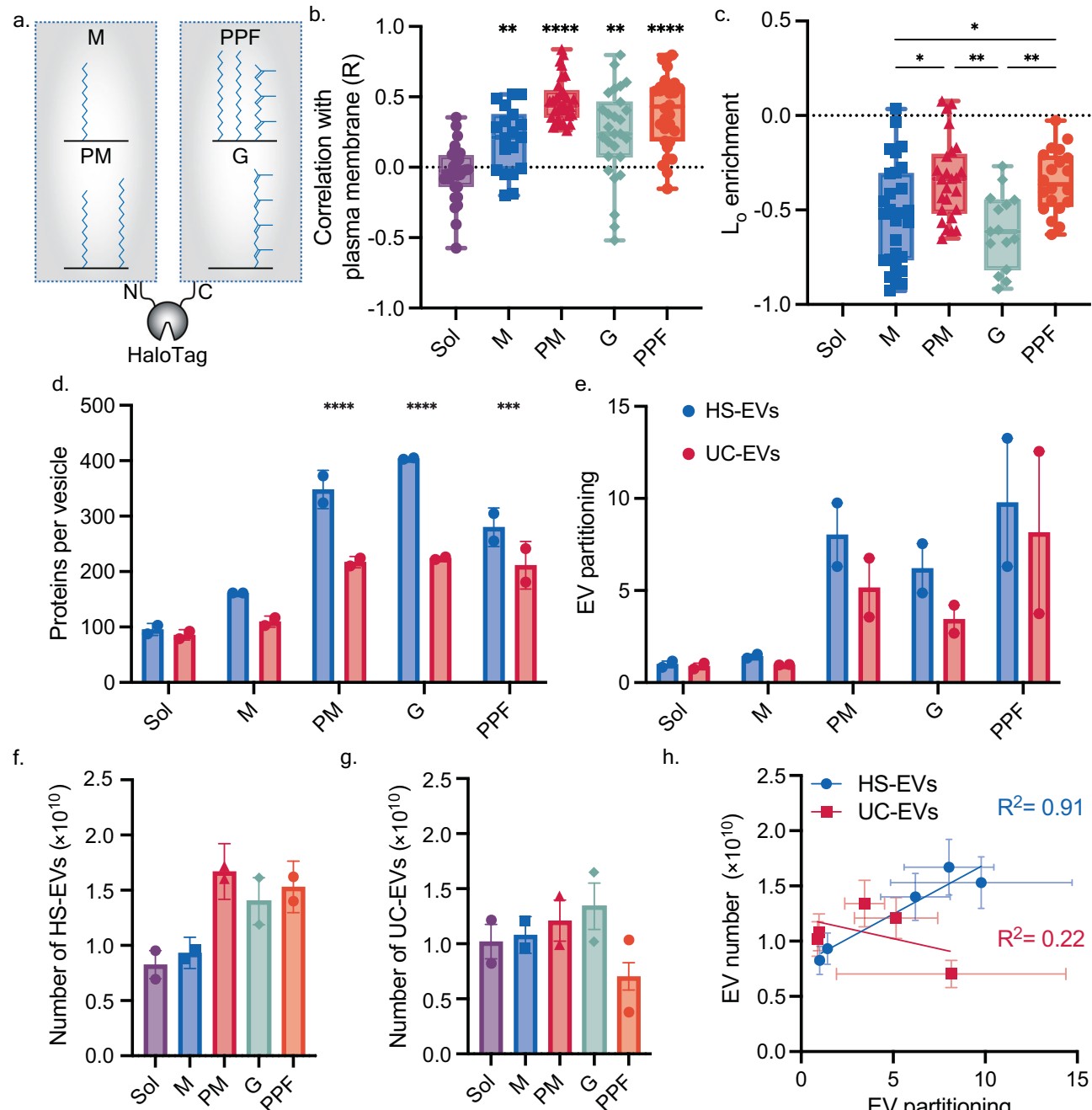

**Fig. 5 | Altering peripheral membrane protein lipidation state affects cellular trafficking, lipid raft association, and extracellular vesicle loading. a** The effect of four lipidation tags on HaloTag membrane interactions and EV loading was assessed. As determined by confocal microscopy, lipidation alters protein association with the **b** plasma membrane and **c** lipid rafts. **b** Pearson's coefficient (R) capturing the covariance between the plasma membrane and the HaloTag signal are plotted for each construct. Data are reported in box and whisker plots collected from 30 cells from two independent experiments; each symbol is a single cell. Welch's one-way ANOVA was performed to compare plasma membrane localization between each lipidated construct and soluble HaloTag, and comparisons were evaluated using Dunnett's T3 multiple comparisons correction (**$p < 0.01$; ****$p < 0.0001$). **c** Reported liquid-ordered ($L_O$) enrichment is the average and standard deviation of at least 15 GPMVs from three independent experiments ($n = 15$). A one-way ANOVA was performed to compare raft localization for each protein, and comparisons were evaluated using Tukey's multiple comparisons correction (*$p < 0.05$; **$p < 0.01$). Comparisons not shown were not significant. For (**b**, **c**), the upper and lower bounds represent the minima and maxima of each data measurement, while the box plot marks the lower and upper quartile, as well as the

median. **d**, **e** Protein loading and partitioning of lipidated HaloTag constructs into EVs as calculated by SDS-PAGE (Supplementary Fig. 17). Each lane of the protein gel received equal numbers of vesicles. A two-way ANOVA (main effects) was performed to compare proteins per vesicle, and comparisons were evaluated using Dunnett's multiple comparisons correction (***$p < 0.001$; ****$p < 0.0001$). Samples were compared to Sol HaloTag. Comparisons not shown were not significant. **e** EV partitioning was defined by normalizing EV protein loading by expression of each construct in EV producer cells as determined by SDS-PAGE. EV partitioning values were normalized by Sol HS-EV partitioning. Lipidation state of HaloTag affects the number of (**f**) HS-EVs generated but not (**g**) UC-EVs, as assessed by NTA (Supplementary Table 2). **h** The number of HS-EVs produced is strongly correlated with EV partitioning of HaloTag constructs. A two-tailed *t*-test was performed to evaluate the significance of the slope when plotting EV partitioning against EV number. The slope for HS-EVs is significantly non-zero ($p = 0.012$), while the slope for UC-EVs was not significantly different than zero ($p > 0.05$). Error bars represent the standard deviation. For (**d**–**h**), $n = 2$ independent sample preparations; error bars represent the SEM. Source data are provided as a Source Data file.

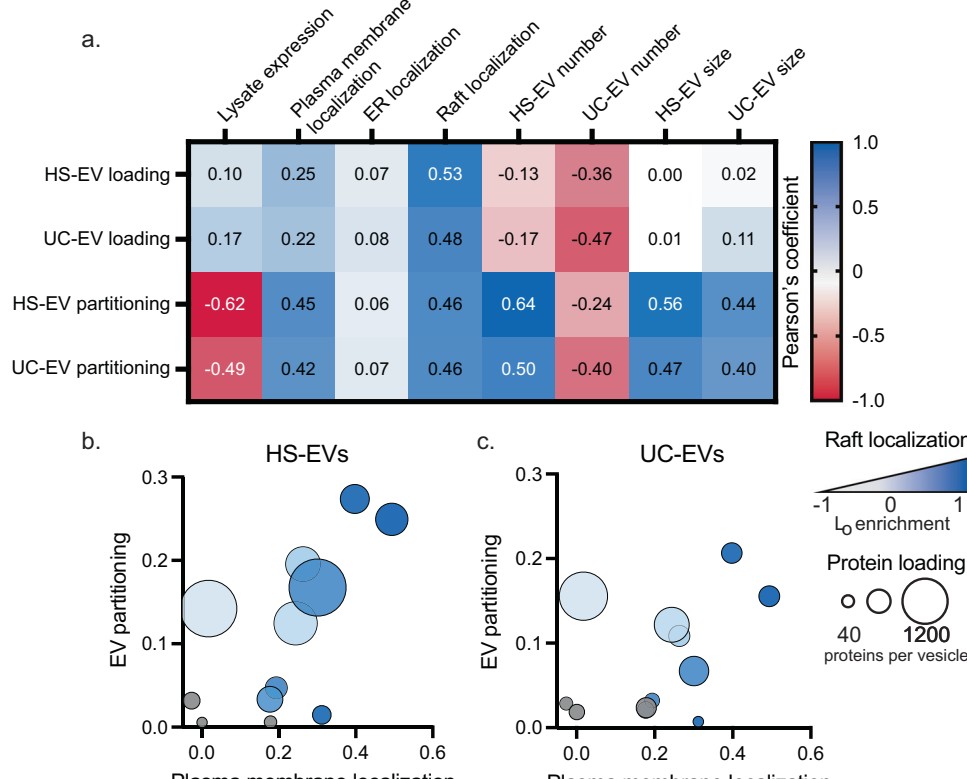

**Fig. 6 | Plasma membrane localization and lipid raft association are the strongest indicators for protein partitioning into EVs. a** Pearson's coefficients for protein loading and partitioning into HS-EVs and UC-EVs for all variables measured. Protein loading (proteins per vesicle) and partitioning (protein loading normalized by total protein expression in EV producer cells) for LAT proteins (LAT WT, LAT C26A, LAT dCore, LAT High ASA), de novo designed proteins (24 Å 4TMD, 32 Å 4TMD, 40 Å 4TMD, 24 Å 12TMD, 40 Å 12TMD), and lipid tagged HaloTag proteins (Sol, M, PM, G, PPF) were compared to protein expression in lysate, localization to the plasma membrane and endoplasmic reticulum (ER), association with lipid rafts, and vesicle number produced and size. *P*-values (two-tailed) for Pearson's coefficients can be found in Supplementary Fig. 19. **b**, **c** Plasma

membrane localization, EV partitioning, raft localization, and protein loading were also plotted in multivariable plots for (**b**) HS-EVs and (**c**) UC-EVs. Plasma membrane localization is calculated as the Pearson's coefficient between plasma membrane stain and each protein; EV partitioning was calculated by normalizing proteins loaded into EVs by expression of each construct in EV producer cells as determined by SDS-PAGE (Figs. 3g, 4e, 5e). Proteins which associated more strongly with lipid rafts are represented in darker blue. Gray points represent constructs for which a raft localization value could not be determined. The size of each point corresponds to the number of proteins per vesicle as calculated by SDS PAGE (Figs. 3f, 4d, 5d). Source data are provided as a Source Data file.

loading into EVs. Specifically, as protein association with lipid rafts is enhanced, loading into EVs generated from these cells is also enhanced. We further synthesized this information by systematically evaluating how HS-EV and UC-EV loading and EV partitioning correlate with the cellular trafficking, lipid interaction, and EV property data (Figs. 3–5, Supplementary Table 2, 3). Plasma membrane localization and raft association correlated strongly with HS-EV and UC-EV partitioning (Fig. 6a). Protein association with lipid rafts has been shown to enhance plasma membrane localization[28]. Because of this relationship, it is unsurprising that both raft and plasma membrane localization correlate with protein loading into EVs, and this observation supports the hypothesis that lipid rafts play a role in EV biogenesis[12]. This relationship is further emphasized when comparing the percentage of EV and non-EV raft proteins which localize to the plasma membrane (Supplementary Fig. 11). The correlation between raft association and EV partitioning was even stronger for the LAT proteins and soluble HaloTag constructs when each family of proteins was analyzed separately (reaching Pearson's coefficients of up to 0.82) (Supplementary Fig. 18). Furthermore, the raft association of palmitoylated proteins is potentially underestimated; DTT was used in the process to form GPMVs, and DTT can cleave disulfides and thioesters leading to depalmitoylation of proteins[16]. Thus, palmitoylated proteins (LAT, PM, and PPF) are potentially more strongly raft associated in cellular systems and more correlated to EV loading than was observed in this assay.

To further visualize protein loading trends, we generated multivariable plots comparing plasma membrane localization, protein partitioning, raft localization, and protein loading into HS-EVs and UC-EVs. From these plots, we observe that plasma membrane and raft localization are positively correlated with protein partitioning into both populations of vesicles (Fig. 6b, c). These observations, derived from data collected on structurally diverse proteins, strongly support the hypothesis that plasma membrane and raft localization are critical parameters for protein partitioning into EVs. Furthermore, the strong agreement between our bioinformatic analysis (Figs. 1d, 2) and EV loading data for a diverse set of natural and de novo designed proteins (Figs. 3–5) suggests that strategies presented here may be applied to improve loading of a wide array of protein classes into EVs.

**An engineered, lipidated transcription factor loads into EVs and drives gene expression after EV-mediated delivery to target cells**
We next investigated whether features that enhance raft association and EV loading could be harnessed to enable engineered EV therapeutics. While simply enhancing EV loading could lead to better protein delivery to target cells, it is also possible that the modification of protein cargo to enhance association with EVs might inhibit the functionality of the cargo or prevent its release from the EV membrane upon membrane fusion with the host cell. Therefore, we next evaluated how modification of a bioactive protein to enhance its loading

into EVs affected its functional delivery and release (into the cytoplasm) in recipient cells.

We chose to explore the EV-mediated, functional delivery of a synthetic transcription factor (synTF) as a model system[42,43]. We utilized the COmposable Mammalian Elements of Transcription (COMET) toolkit that comprises a set of synTFs and cognate promoters[43]. We fused lipidation tags to a synTF, which consists of a zinc-finger DNA binding domain, a p65 transcriptional activator domain, and a nuclear localization sequence, and then we assessed cargo loading and functional delivery to a HEK293FT-based reporter cell line (Fig. 7a, Supplementary Fig. 20). This reporter line contains a genomically integrated expression cassette that includes the cognate promoter for our selected synTF that regulates a dsRed-Express2 reporter gene; functional delivery of a synTF is indicated by induction of dsRed-Express2 fluorescence. In addition, the expression cassette contains a constitutive miRFP720 fluorescent protein to identify cells with an active reporter locus.

To determine if lipidated synTFs were capable of driving reporter expression, we first directly transfected synTF-encoding plasmids into reporter cells. The addition of lipid tags to the transcription factor impaired the ability to induce reporter expression to different extents across our constructs (compared to the non-lipidated "Sol" synTF control), although all transcription factors were still functional to some degree (Fig. 7b). Differences in synTF-induced reporter expression were not attributable to differential expression of the synTFs, suggesting that lipidation modulates specific (per protein) functional activity of the synTFs (Fig. 7b, Supplementary Fig. 21). The differences in functionality between lipidated synTFs may be partially attributed to lipidation-mediated sequestration of synTFs to a membrane, preventing them from entering the nucleus. We next harvested EVs from cells transfected with synTF plasmids and determined that the more raft associated tags, PM and PPF (Fig. 5c), led to the best loading of synTFs into EVs (Fig. 7c, Supplementary Fig. 21). Most notably, the PM and PPF tags increased EV loading relative to the soluble synTF by 62X and 240X, respectively. By semi-quantitative western blot, these loading increases corresponded to loading 54 or 180 synTFs/EV for PM and PPF, respectively (Supplementary Fig. 21). Taking the EV-loading data from lipidated HaloTag and synTF together, we conclude that lipid raft association is an EV-loading strategy that is robust to protein cargo identity. This contrasts with strategies for enhancing loading that are not raft-dependent; non-raft tags such as G conferred loading of the HaloTag construct but not the synTF construct (Fig. 5d, e, Fig. 7c).

Finally, we evaluated whether EV-mediated delivery of lipid-modified synTFs could drive functional changes in recipient cells. We treated reporter cells with EVs containing lipidated synTFs and a surface displayed viral glycoprotein (vesicular stomatitis virus G protein, VSV-G) to promote EV uptake and membrane fusion with recipient endosomal membranes[44]. Notably, EV-mediated delivery of several lipidated synTF designs successfully induced reporter expression, while EV-mediated delivery of synTF (Sol) conferred no substantial reporter induction (Fig. 7d–f, Supplementary Fig. 22). Small differences in VSV-G loading into EVs were observed across the SynTF variants (Supplementary Fig. 23); these minor variations are unlikely to explain the large differences in functional delivery observed in Fig. 7f. The best performing design, PM-modified synTF, partitions strongly to lipid rafts (Fig. 5c) and converted up to ~7% of reporter cells to a substantially activated state (Fig. 7f). These observations highlight the utility and tradeoffs associated with using a lipid raft associating tag such as PM: despite having <10% of the specific (per protein) transcriptional activity relative to a soluble synTF (Fig. 7b), the PM tag enabled EV-mediated delivery of synTF cargo to an extent that far exceeded the soluble synTF. We speculate that discrepancies between EV loading and functional delivery for some lipid tags (i.e. PPF's high loading but low functional delivery) may be due to a differential ability

of the tags to dissociate from the EV membrane after delivery and membrane fusion with reporter cells. Presumably, lipid tags that enable association during EV generation but disassociation after delivery into target cells should yield the most effective modifications for TF delivery. It is also likely that tags which drive EV loading may lead to a loss in apparent specific activity in a high synTF expression assay (e.g., Fig. 7b), but this effect is countered by the benefits of active loading and/or may not diminish overall function if the synTF eventually dissociates and drives target gene activity in the recipient cells. These subtleties highlight the value of a "balanced" tag such as PM and indicate opportunities for future exploration towards applying principles elucidated here for EV engineering.

## Discussion

In this study, we showed that protein association with the plasma membrane and lipid rafts can be used as a handle to enhance protein loading into extracellular vesicles. Using spectroscopy and bioinformatic analysis, we provide evidence that EVs largely mirror lipid rafts in lipid and protein composition. We then analyzed native proteins which associate with EVs to identify features which may facilitate protein cargo loading into EVs. Based on this analysis, we employed rational protein design to build a library of transmembrane and peripheral membrane proteins with features that we hypothesized would affect membrane interactions and subsequent EV loading. We found that proteins which localized to the plasma membrane and to lipid rafts generally loaded well into extracellular vesicles. Finally, to evaluate the potential utility of exploiting these principles for applications such as engineering EVs to deliver active cargo, we altered the lipid raft association of an engineered transcription factor to drive loading into EVs and functional delivery to target cells to trigger changes in gene expression. Altogether, this work builds fundamental understanding as to how protein loading into EVs can be engineered via modulating membrane-protein interactions.

Our proposed relationship between raft association and EV loading is consistent with other studies in distinct systems that probed the impact of lipidation on protein loading into EVs. For example, Ye et al. investigated how lipid raft disruption by Filipin III treatment modulated loading of two natively expressed (not introduced by transfection) peripheral proteins, Fyn and Src kinase, into EVs isolated from PC-3 cells[45]. They found that Filipin III treatment disrupted EV-loading of Fyn kinase, which possesses a myristic acid and 2 palmitic acids on its N-terminus, but this disruption was not observed for Src kinase, which possesses a single N-terminal myristic acid (the same terminus as our M tag). Based on these results, one would hypothesize that Fyn is raft associated, while Src kinase is not. Our observations support that conclusion; we evaluated the Src kinase tag and observed that it does not mediate association with lipid rafts in GPMVs. In other supporting evidence, the N-terminus of Fyn kinase has been shown to drive raft association in GPMVs[27]. Overall, our results are concordant with these prior studies and extend our understanding of the relationship between lipid-protein interactions and trafficking, EV loading, and functional delivery of proteins.

We speculate that engineering protein-lipid interactions will be a useful strategy in future applications such as loading therapeutic cargo into EVs or generating complex, multifunctional EVs. The addition of transmembrane domains or lipidation sites can effectively load protein cargo into EVs[8,9,46–50], but to the best of our knowledge how and why specific membrane linkages lead to increased protein loading into EVs has not been characterized until now. Because we leverage general biophysical features, derived from the analysis of all human membrane proteins, and demonstrate the trafficking and loading of multiple protein cargos, we envision that this strategy can be applied to improve the loading of diverse protein cargoes. For example, one could imagine a goal of delivering a therapeutic, soluble protein (i.e., CRISPR-Cas9) to a specific cell type. Using the design principles

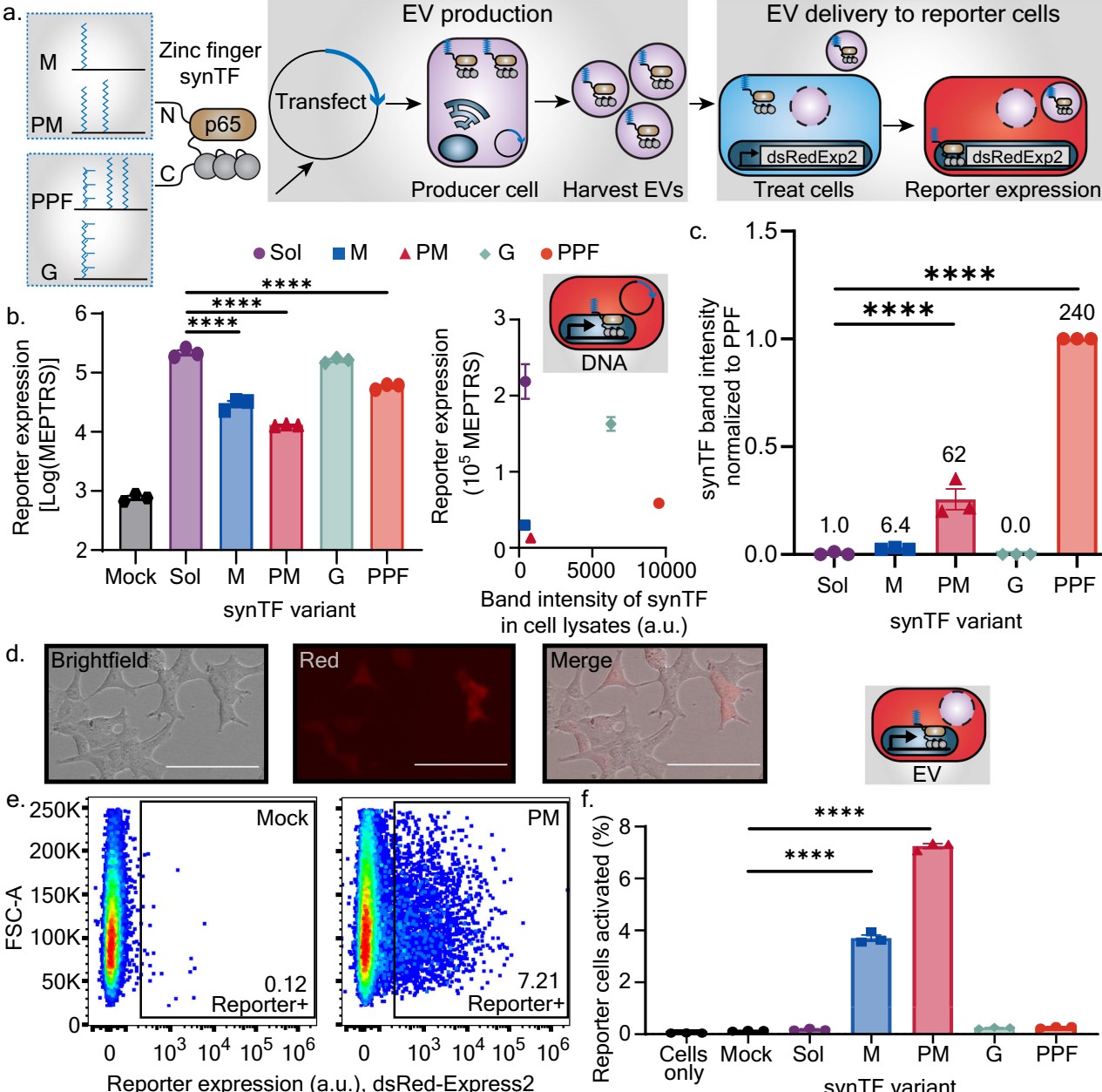

**Fig. 7 | Engineered lipidation of transcription factors enables loading into EVs and functional delivery to recipient cells. a** Schematic of workflow for evaluating the loading into EVs and functional activity of lipidated synthetic transcription factors (synTFs). Reporter cells express a red fluorescent protein, dsRed-Express2 (dsRedExp2), when activated. **b** Activity of synTF variants when plasmids were directly transfected into reporter cells. Left, log transform of reporter expression measured by flow cytometry in absolute units of mean molecules of equivalent phycoerythrin-texas red (PE-TR) (MEPTRs). Each symbol is an independent biological replicate, and error bars are the SEM ($n = 3$). A one-way ANOVA was performed, and comparisons were evaluated using Dunnett's multiple comparisons correction (****$p < 0.0001$). synTFs were compared to the Sol condition; comparisons not shown were not significant. Right, scatter plot for the experiment on the left plotted against band intensity of synTF expression in those cells as measured by a single western blot ($n = 1$) (Supplementary Fig. 21). **c** Western blot band intensities of synTFs loaded into EVs, normalized to each individual blot's PPF condition. Each symbol represents a biologically independent EV preparation and western blot, and the error bars represent SEM ($n = 3$). The numbers above each bar are the average fold increase of synTF loading compared to the Sol synTF condition. A one-way

ANOVA was performed, and comparisons were evaluated using Dunnett's multiple comparisons correction (****$p < 0.0001$). SynTFs were compared to the Sol condition; comparisons not shown were not significant. **d** Representative micrograph of reporter cells activated via EV-mediated delivery of lipidated synTFs (here, the PM variant). Scale bar is 100 μm. **e** Flow cytometry dot plots for a representative sample of reporter cells treated with either EVs isolated from Mock-transfected cells (no synTF) or from cells expressing PM-tagged synTF. Annotations state the percentage of reporter cells that expressed dsRedExp2. **f** Activation of reporter cells treated with synTF-containing EVs (evaluated by flow cytometry). Each symbol represents an independent biological replicate, and error bars represent SEM ($n = 3$). A one-way ANOVA was performed, and comparisons were evaluated using Dunnett's multiple comparisons correction (****$p < 0.0001$). SynTFs were compared to the Mock condition; comparisons not shown were not significant. Throughout the figure, "Mock" refers to conditions where an empty backbone plasmid was transfected into a cell in lieu of a SynTF-encoding plasmid. Data in (**c**, **e**, and **f**) are representative of two independent experiments. Data in panel (**b**) were collected from one experiment. Source data are provided as a Source Data file.

developed herein, one could load cargo by employing raft-associating lipidation tags, and load one or more surface displayed, transmembrane-tethered single chain fragment variables (scFvs) that bind to surface markers of the desired target cell type[51]. We speculate that the EV-loading of each protein species may be independently tuned via selection of lipid tag composition or modification of protein transmembrane physical features. In general, we anticipate that harnessing membrane-protein interactions as a tool will be compatible with other post-harvest or genetic engineering-based EV loading strategies.

The functional delivery of lipidated synTFs demonstrated here extends the known design principles for loading epigenetic-modifying cargo into EVs. Several groups have delivered synthetic transcription regulators via EVs to control genetic programs[44,52–55]; our work complements these technological efforts by demonstrating that several lipid tags (M, PM) fused directly to genetic cargo can alone enable EV-mediated delivery. Others have speculated that tight plasma membrane association, such as that conferred by a raft-associating, palmitoylation-based tag such as PM, would be undesirable for an EV-delivered, nucleus-bound cargo[56]. We were similarly surprised that PM could drive reporter expression given its strong raft and membrane interactions; yet PM was our highest performing tag in EV-mediated functional delivery studies. Interestingly, the PPF tag conferred the highest loading of protein cargo, and it performed better than the PM tag when directly expressed in reporter cells but did not enable functional delivery of our synTF. Presently, we cannot explain the discrepancy between loading and EV-mediated activity (i.e., reporter gene expression) between the PM and PPF tags and believe this is an interesting area for future research. The difference in functional delivery may possibly arise from the ability of each tag to enable transport to the nucleus. One potential hypothesis is that the PPF-tagged synTFs may be more tightly bound to membranes than PM-tagged synTFs in reporter cells because the PPF tag is composed of 3 lipids (2 palmitic acids and 1 farnesyl) compared to two for PM (1 myristic and 1 palmitic acid)[57]. Alternatively, differences in delivery could be explained by inherent trafficking differences between lipidation tags−indeed, the SH4 domain of Lyn, the protein domain from which the PM tag is derived, drives association of Lyn with the plasma membrane and internal membranes such as the Golgi and nuclear membranes[58–60]. Tight membrane association of soluble cargo may be useful in some applications, such as disrupting membrane-proximal signaling events. Together, these results highlight how the desired function post-delivery, in addition to loading and trafficking into EVs, must be considered when choosing a protein loading strategy.

Beyond applications in EV engineering, this work demonstrates how membrane biophysical features, such as membrane fluidity, lipid order, and organization, can be harnessed to enhance the capabilities of engineered cellular systems. In this work, we demonstrate how altering protein trafficking through the modulation of lipid-protein interactions can be used to tune protein loading into EVs. Biophysical interactions between lipids and proteins also alter protein trafficking, lateral organization, protein-protein interactions, and activity of proteins within lipid membranes[25,27,28,34,61]. Viewing membrane biophysics as a tool for cellular engineering could lead to improved performance of membrane-based therapeutics and biosensors[61,62]. For example, synthetic cell surface receptors that drive engineered cell behavior in response to detecting ligands are key components of many modern cell therapies[63–65]. Some receptor designs are limited by poor plasma membrane trafficking[64], which could be overcome by increasing transmembrane domain length or adding lipidation tags (Fig. 3). Synthetic receptors often rely on interactions with other membrane proteins to drive intracellular signaling; these interactions should allow us to tune the performance of these receptors by localizing such synthetic receptors to or away from such signaling proteins via

membrane-protein interactions (e.g., to raft or non-raft regions). Ultimately, our insights suggest that engineering protein-lipid interactions is a useful design consideration when developing protein-based cellular systems, which we hope will be useful for developing EV and cell-based therapies.

## Methods

### Bioinformatics

To perform the bioinformatic analysis, we first generated a list of raft proteins populating the list of human raft associated proteins in RaftProt 2.0[23] with protein sequence and structural information from Swiss-Prot. Similarly, we created a database of human EV proteins by pulling protein information from Swiss-Prot for proteins found in Exocarta[22], an EV database. To determine the probability of a raft protein or membrane-associated protein to be found in an EV, 100 human proteins were randomly selected from either RaftProt 2.0 or a list of all human membrane-associated proteins in Swiss-Prot, and the fraction of each subset of proteins found in Exocarta was calculated, reported as the frequency found in EVs (Fig. 1d). By utilizing our database of annotated raft, EV, and membrane associated proteins, we determined the average transmembrane domain length, frequency and number of lipidation sites (myristoylation, palmitoylation, prenylation), glycosylation, phosphorylation, and the presence of disulfide bonds. Transmembrane domain lengths were estimated by multiplying the number of amino acids within the transmembrane domains by 0.15 nm[24,25]. Transmembrane domain length for multi-pass transmembrane domain proteins is reported as the average length for all transmembrane domains. The number of proteins in each category are listed in Supplementary Table 1.

### Cloning and DNA preparation

Plasmid cloning was performed using standard PCR and restriction enzyme cloning techniques. Genes were typically ordered as gBlocks from Twist Biosciences, and primers for cloning were ordered from Integrated DNA Technologies. Genes encoding proteins used for protein localization were cloned into the Twist Biosciences CMV mammalian expression vector (CMV Puro). A gBlock encoding the p65 ZF6 synTF was ordered from Twist and cloned into plasmids with lipidation tags. Plasmids encoding fluorescent proteins were generally cloned into a modified pcDNA3.1 backbone (Thermo Fisher #87020b). Plasmid maps are available in Supporting Information 1; DNA sequences and sources of key sequences are in Supplementary Table 4-6. Additionally, plasmids have been deposited to Addgene (Addgene plasmids #202492-202510). The BxB1 recombinase expression vector plasmid was a gift from Ron Weiss. DsRed-Express2 was obtained by site-directed mutagenesis of pDsRed2-N1, which was a gift from David Schaffer (University of California, Berkeley). BlastR was sourced from lenti dCAS-VP64_Blast, which was a gift from Feng Zhang (Addgene plasmid #61425)[66]. The p65 sequence was sourced from SP-dCas9-VPR, which was a gift from George Church (Addgene plasmid # 63798)[67]. The mutated LAT sequences were from Ilya Levental[25]. The M lipid tag sequence is from William Rogers[68] and the PM and G lipid tag sequences were from Barbara Baird and David Holowka[69]. The PPF lipid tag is from Paralemmin-1 (O75781-1). The insulator for the synTF reporter constructs came from PhiC31-Neo-ins-5xTetO-pEF-H2B-Citrin-ins, which was a gift from Michael Elowitz (Addgene plasmid #78099)[70]. The EBFP2 plasmid[43,71] and the miRFP720[72] sequence have been previously described. The plasmid encoding VSV-G, pMD2.G, was a gift from William Miller. Enzymes and buffers required for PCR and cloning were purchased from Thermo Fisher and/or NEB. Plasmid DNA was prepared by growing 100 mL of Top10 *E. coli* overnight with the appropriate selective antibiotic and then purifying using the PureLink Plasmid Midiprep Kit (Thermo Fisher), ZymoPURE II Midiprep Kit (Zymo Research) or polyethylene glycol precipitation[43].

### Golden gate assembly of synTF reporter integration vector

The synTF (specifically, ZF6[43]) reporter integration vector was assembled as a landing pad-based integration vector through a BbsI-mediated Golden Gate reaction[73]. Each 20 μL reaction comprised 2 μL 10 X T4 ligase buffer, 2 μL 10X BSA (1 mg/mL stock), 0.8 μL BbsI-HF restriction enzyme (NEB #R3539L), 0.8 μL T4 DNA Ligase (400 U/μL stock), 20 fmol integration vector backbone and 40 fmol of each transcription unit and linker plasmid to be inserted. The backbone used here was pPD1178, which contains a promoter-less expression unit of the puromycin resistance gene and miRFP720 connected via P2A; both genes are expressed upon successful integration in the landing pad downstream of the landing pad's CAG promoter[73]. The reaction also included six transcription units of the synTF (ZF6) promoter with 12 compact binding sites upstream of a YB_TATA minimal promoter[74,75] and a DsRed-Express2 reporter gene (pHIE324−329), a seventh transcription unit containing a constitutive hEF1a promoter and a blasticidin resistance gene (pHIE280), and a linker plasmid to close the assembly (pPD1157, Addgene #139245). Transcription unit vectors for each reporter copy and the linker vector were previously described[43].The reaction was incubated at 37 °C for 15 min, then subjected to 55 iterations of thermocycling (37 °C for 5 min, 16 °C for 3 min, repeat), followed by 37 °C for 15 min, 50 °C for 5 min, 80 °C for 10 min to terminate the reactions; then the mixture was cooled to room temperature ( ~ 22 °C) prior to immediate transformation into NEB Stable chemically competent bacteria (NEB #C3040H).

### Cell culture

The HEK293FT cell line was purchased from Thermo Fisher/Life Technologies. The HEK293LP cell line was a gift from Ron Weiss[73]. Cells were cultured in Dulbecco's Modified Eagle Medium (DMEM) (Gibco 31600-091) with additional 3.5 g/L glucose (Sigma G7021), 3.7 g/L sodium bicarbonate (Fisher S233), 10% FBS (Gibco 16140-071), 6 mM L-glutamine (2 mM from Gibco 31600-091 and 4 mM from additional Gibco 25030-081), penicillin (100 U/μL), and streptomycin (100 μg/mL) (Gibco 15140122), in a 37 °C incubator with 5% $CO_2$. Prior to flow cytometry and for certain microscopy studies, cells were cultured temporarily in phenol red-free DMEM from Sigma (D2902). This base medium was supplemented with 4 mg/L pyridoxine-HCl (Sigma P6280), 16 mg/L sodium phosphate (Sigma S5011), 3.7 g/L sodium bicarbonate, 3.5 g/L glucose, 100 U/mL penicillin, 100 μg/mL streptomycin, 4 mM L-glutamine, and 10% FBS.

### Laurdan spectroscopy and liposome formation

Liquid disordered membranes were composed of 70 mol% 1,2-dio-leoyl-sn-glycero-3-phosphocholine (DOPC) and 30 mol% cholesterol, while liquid ordered membranes were composed of 70 mol% 1,2-dipalmitoyl-sn-glycero-3-phosphocholine (DPPC) and 30 mol% cholesterol. Lipids were purchased from Avanti. Liposomes were prepared via the thin film hydration method. Lipids dissolved in chloroform were dried under nitrogen gas in a glass vial and placed in a vacuum oven for at least 2 h. Films were rehydrated with PBS at 60 °C for at least 2 h before being vortexed and extruded (21 passes) through a 100 nm polycarbonate filter (Avanti Mini Extruder).

C-Laurdan was used to measure the membrane fluidity of cell, GPMV, EV, and liposome membranes. C-Laurdan was dissolved in DMSO at 20 mM and diluted to 8 μM in PBS as an intermediate stock. 100 μM of lipid was used for Laurdan measurements of liposomes. Approximately, $1 \times 10^9$ HS-EVs and UC-EVs, $1 \times 10^6$ cells, and GPMVs formed from $1 \times 10^6$ cells were used in each measurement. For bulk measurements, 0.4 μM C-Laurdan (Tocris, 7273) was added to the sample of interest and incubated in the dark at room temperature for 30 min. C-Laurdan spectra was then read (ex. 350 nm) using a Molecular Devices Spectra Max i3 plate reader (SoftMax Pro 7.1). Membrane hydration was analyzed by calculating Laurdan generalized

fluorescence polarization (GP) through the following formula:

$$GP = (I_{439} - I_{483})/(I_{439} + I_{483}) \qquad (1)$$

Where $I_{439}$ and $I_{483}$ are the fluorescence intensities of the sample at 439 and 483 nm, respectively, when excited at 350 nm.

### Transient transfection

Transient transfection of HEK293FT cells was achieved using the calcium phosphate method, described previously[43]. Briefly, cells were plated in DMEM and allowed to adhere to the plate for 5–8 h. Plasmids encoding the protein of interest were diluted in a water and $CaCl_2$ solution (0.3 M final), added to an equal volume of 2 x HEPES Buffered Saline (280 mM NaCl, 50 mM HEPES, 1.5 mM $Na_2HPO_4$), mixed 4 times, and incubated for 2–4 min. The DNA solution was then vigorously pipetted and added dropwise to plated cells. Plating density and DNA amounts added for each experiment and plate format can be found in the sections below. The next morning, medium was aspirated and replaced with fresh DMEM. Cells were cultured for at least an additional 24 h before performing downstream experiments.

### Generation of the HEK293FT synTF (ZF6) reporter cell line (HIE156)

From exponentially growing HEK293LP cells[73], $0.5 \times 10^5$ cells were plated per well (0.5 mL medium) in 24-well format, and cells were cultured for 24 h to allow cells to attach and spread. The BxB1 recombinase variant used is a mammalian codon optimized recombinase with a C-terminal NLS[73]. BxB1 recombinase was co-transfected with the integration vector by lipofection with Lipofectamine LTX with PLUS Reagent (Thermo Fisher 15338100). 300 ng of BxB1 expression vector was mixed with 300 ng of integration vector, 0.5 μL of PLUS reagent, and enough OptiMEM (Thermo Fisher/Gibco 31985062) to bring the mix volume up to 25 μL. In a separate tube, 1.9 μL of LTX reagent was mixed with 23.1 μL of OptiMEM. The DNA/PLUS reagent mix was added to the LTX mix, pipetted up and down four times, and then incubated at room temperature for 5 min. 50 μL of this transfection mix was added dropwise to each well of cells and mixed by gentle swirling. Cells were cultured until the well was ready to split (typically 3 d), without any media changes.

To begin selection of cells that successfully integrated the synTF reporter integration vector, cells were harvested from the 24-well plate when confluent by trypsinizing and transferring to a single well of a 6-well plate in 2 mL of medium supplemented with 1 μg/mL puromycin (Invivogen ant-pr). Cells were trypsinized daily (typically 3 d) until cell death was no longer evident. Cells were cultured in medium supplemented with puromycin until the 6-well was confluent and cells were exponentially growing. Cells were then selected with 6 μg/mL blasticidin (Alfa Aesar/Thermo Fisher J61883) for 7 d. Cells were cultured in both puromycin and blasticidin to maintain selective pressure until flow sorting.

To sort, cells were harvested by trypsinizing, resuspended at -$10^7$ cells per mL in pre-sort medium (DMEM with 10% FBS, 25 mM HEPES (Sigma H3375), and 100 μg/mL gentamycin (Amresco 0304)), and held on ice until sorting was performed. Cells were sorted using a BD FACS Aria 4-laser Special Order Research Product (Robert H. Lurie Cancer Center Flow Cytometry Core). The sorting strategy was as follows: single cells were first gated to exclude all EYFP positive cells (as EYFP positive cells still have an intact landing pad locus, suggesting a mis-integration event occurred) and to include only miRFP720+ cells. EYFP expression was measured using the FITC channel (488 nm excitation laser, 505 LP and 525/30 emission filters) and miRFP720 expression was measured using a modified APC-Cy7 channel (640 nm excitation laser, 690 LP and 730/45 emission filters). Then a gate was drawn on miRFP720 expression to capture the 88th–98th percentile of miRFP720-expressing cells (the top 2% were excluded to exclude cells

suspected to possess two or more integrated copies of the cargo vector). Fifty thousand cells were collected in post-sort medium (DMEM with 20% FBS, 25 mM HEPES, and 100 µg/mL gentamicin), and cells were held on ice until they could be centrifuged at 150 $g$ for 5 min, resuspended in 0.5 mL complete medium supplemented with 100 µg/mL gentamicin, and plated in one well of a 24-well plate. Cells were maintained in gentamicin for 7 days after sorting during expansion before banking. Cells were thawed for use in experiments in this study.

### Protein trafficking analyzed via live cell imaging
Protein localization to the plasma membrane and endoplasmic reticulum was analyzed in live cells using confocal microscopy. To facilitate strong cell attachment, 24-well plates were coated with poly-L-lysine (Sigma P6282, resuspended at 0.1 mg/mL in sterile water) by incubating wells with poly-L-lysine for at least 5 min at room temperature before aspirating excess solution and allowing the plates to dry. $1 \times 10^5$ HEK293FTs were plated into treated wells and transfected with 20 ng of DNA encoding transmembrane proteins and 60 ng of DNA encoding for peripheral membrane proteins. Forty-eight hours after transfection, cells were imaged. All proteins were fused to either RFP or HaloTag. For proteins fused to HaloTag, proteins were tagged with TMR ligand for visualization. TMR Ligand (Promega) was diluted 1:200 in PBS and used as a 5 × stock. One-fifth of the existing volume of cell medium was replaced with the TMR HaloTag solution (5 µM TMR final concentration) and incubated for 15 min in a 37 °C and 5% $CO_2$ cell culture incubator. The ligand containing medium was then gently washed with fresh medium and replaced with 500 µL of fresh medium containing 5 µM ER Tracker Blue-White DPX (Thermo Fisher) and 1:1000 Cell-Mask Deep Red (Thermo Fisher). Cells were then incubated at 37 °C and 5% CO2 in a cell culture incubator for 30 min. Medium was then replaced with phenol red-free DMEM. Once labeled, cells were imaged using a Nikon confocal microscope. A 561 nm laser was used to excite the HaloTag- or mRFP1-labeled proteins, a 405 nm laser was used to excite the endoplasmic reticulum dye (ER Tracker Blue-White DPX), and a 640 nm laser was used to excite the plasma membrane dye (Cell-Mask Deep Red). To observe the entire cell, Z-stacks were captured and converted into maximum projection image prior to analysis. Correlation of protein localization with the plasma membrane and endoplasmic reticulum was calculated using the Nikon NIS Elements software (AR 5.21.03) for whole cells.

### Measuring protein raft localization in GPMVs
Protein association with lipid rafts was measured using GPMVs. Approximately $8 \times 10^5$ HEK293FTs were plated into 6-well plates and transfected with 80 ng of DNA encoding transmembrane proteins and 240 ng of DNA encoding for peripheral membrane proteins. 48 h after transfection, GPMVs were produced. All proteins were fused to either RFP or HaloTag. For proteins fused to HaloTag, proteins were tagged with TMR ligand for visualization. TMR Ligand (Promega) was diluted 1:200 in PBS and used as a 10 × stock. One-tenth of the existing volume of cell medium was replaced with the TMR HaloTag solution (2.5 µM TMR final concentration) and incubated for 15 min in a 37 °C and 5% $CO_2$ cell culture incubator. Cells were then washed with GPMV buffer (5 mM HEPES, 75 mM NaCl, 1 mM CaCl2, pH 7.4) and then incubated for 1 h in a 37 °C and 5% $CO_2$ cell culture incubator in GPMV vesiculation buffer (GPMV buffer with 0.08% PFA and 2 mM DTT). GPMVs were then carefully collected, making sure to not disturb the cells, and allowed to sink in a fresh tube. Prior to imaging, GPMVs were labeled with 5 µg/mL of Fast DiO. Samples were then placed onto a glass slide and cooled to 10 °C on the microscope stage using a Linkam PE100 Peltier Stage. Once the sample temperature reached 10 °C, GPMVs were imaged using a Nikon confocal microscope. A 561 nm laser was used to excite the protein label, and a 488 nm laser was used to excite the membrane dye. Line scans were performed to measure the protein fluorescence in raft ($I_{Raft}$) and nonraft ($I_{Non\ Raft}$) regions, as marked by DiO. Protein

partitioning ($L_o$ enrichment) in lipid rafts was calculated using the Nikon NIS Elements Software.

$L_o$ Enrichment was calculated using the following equation:

$$L_O\ Enrichment = \frac{I_{Raft} - I_{Non\ Raft}}{I_{Raft} + I_{Non\ Raft}} \qquad (2)$$

Where $I_{Raft}$ represents the protein fluorescence intensity in the raft phase, and $I_{Non\ Raft}$ represents the protein fluorescence intensity in the nonraft phase.

### EV production and isolation
$5 \times 10^6$ HEK293FTs were plated in 10 cm tissue culture treated plates in 8-10 mL DMEM. 5-8 h later, cells were transfected as described above ("Transient transfection"). Total DNA amount was between 5 and 20 µg per plate and kept consistent for all conditions in a given experiment. The typical amount of plasmid encoding the protein of interest was 2–5 µg per plate. The remainder of plasmid was empty vector (pcDNA) and for EV transfections with HaloTag- or synTF-containing constructs (i.e., Fig. 3f, g, Fig. 5d, e, Fig. 7c) a blue transfection marker (<0.5 µg of eBFP2). The next morning after transfection, medium was replaced with DMEM supplemented with 10% of EV-depleted FBS (Gibco A2720801 or Omega Scientific FB40/50). Supernatant was harvested for EV isolation ~30–34 h later and clarified by sequential centrifugation at 300 g for 10 min and 2000 g for 20 min. HS-EVs were pelleted by a Beckman Coulter Avanti J-26XP centrifuge with a J-LITE JLA 16.25 rotor at 15,000 g for 30 min. UC-EVs were pelleted by centrifuging the supernatant at 120,416 g for 135 min in polypropylene ultracentrifuge tubes in a Beckman Coulter Optima L-80 XP model and a SW 41 Ti rotor. EVs were resuspended in the conditioned cell medium remaining in their tubes via gentle pipetting. Each centrifugation step was performed at 4 °C, and samples were stored on ice for short-term use or at -80 °C for long-term storage.

### Nanoparticle tracking analysis and EV production measurements
EV concentration and size were measured via a NanoSight NS300 (Malvern) instrument with a 642 nm laser on software v3.4. EVs were diluted in PBS to $2-10 \times 10^8$ particles/mL, infused at setting 30, imaged with camera level 14, and analyzed at detection threshold 7. Data reported are averaged from the analysis of three 30 s videos per sample. For data presented in Figs. 3–5, the number of EVs produced for a given experiment were collected from two 10 cm plates of transfected HEK293FTs.

### Cell lysate isolation and BCA
Cell lysates were harvested from HEK293FTs using radio-immunoprecipitation assay buffer (RIPA); samples were collected at the same time as EV supernatants. Briefly, the medium was removed, the cells were washed 2 × in ice cold PBS and then lysed with ice cold RIPA (150 mM NaCl, 50 mM Tris-HCl pH 8.0, 1% Triton X-100, 0.5% sodium deoxycholate, 0.1% sodium dodecyl sulfate, and protease inhibitor (Pierce #A32953)). Lysates were transferred to cold microcentrifuge tubes and incubated for at least 30 min on ice. Samples were clarified by centrifugation at 14,000 g x 20 min at 4 °C, and the supernatant was transferred to fresh tubes. Protein concentration was determined via bicinchoninic acid (BCA) assay (Pierce #23225), and samples were kept on ice for short-term storage or frozen at -80 °C for long-term storage.

### Quantification of protein loading into EVs
Protein loading into EVs was quantified either via fluorescent PAGE gel or western blot. For protein constructs fused to HaloTag, 17 µL of cell lysates or EVs diluted in PBS were incubated with 1 µL of 1:100 TMR ligand at room temperature for 15 min. Laemmli buffer (without

bromophenol blue for HaloTag gels) was then added and samples were heated to 70 °C for 3 min. Approximately $5 \times 10^8$ vesicles and purified HaloTag standard curve were loaded and run on a 12% Mini-PROTEAN TGX Precast Protein Gel (Bio-Rad) at 150 V for 90 min or 50 V for 10 min followed by 100 V for ~70 min. 5 μg of cell lysate for each construct was run on a separate gel. Once complete, gels were washed in Milli-Q water and imaged using an Azure Sapphire Imager (Azure cSeries Acquisition software v1.9.5.0606). Gels were analyzed using ImageJ (Fiji 3[76]).

For protein constructs without HaloTag, PAGE gels were run in the same manner, except for the HaloTag ligand incubation step. Additionally, purified FLAG tagged protein (recombinant p53 protein, R&D Systems) was run with vesicles to quantify loading. Wet transfer was performed onto a PVDF membrane (Bio-Rad) for 45 min at 100 V. Membranes were then blocked for an hour at room temperature in 5% milk in TBST (pH 7.6: 50 mM Tris, 150 mM NaCl, HCl to pH 7.6, 0.1% Tween 20) and incubated for 1 h at room temperature or overnight at 4 °C with primary solution (anti-FLAG (Sigma F1804), diluted 1:1000 in 5% milk in TBST). Primary antibody solution was decanted, and the membrane was washed three times for 5 min in TBST and then incubated in secondary solution at room temperature for 1 h (HRP-anti-Mouse (CST 7076) diluted 1:3000 in 5% milk in TBST). Membranes were then washed in TBST and incubated with Clarity Western ECL Substrate (Bio-Rad) for 5 min. Blots were imaged using an Azure Biosystems c280 or c600 imager, and band intensities were quantified with ImageJ. Exceptions to the above protocol are listed in Supplementary Table 7.

### EV-mediated synTF delivery experiments

To generate synTF-containing EVs, HEK293FTs were transfected as described above ("EV production and isolation") with plasmids encoding the following: synTF variant (~5 μg), vesicular stomatitis virus G (VSV-G) viral fusion protein (3 μg), blue color control (eBFP2) (0.3–0.5 μg), and an empty-pcDNA3.1 backbone (to bring the total DNA transfected up to 10 μg per plate). Because each synTF variant is slightly different in plasmid size, cells were transfected such that each plate received equal copy numbers of plasmids, generally around 5 μg. The Mock EV conditions were transfected as per above but with the DNA mass of the synTF variant replaced with additional empty pcDNA3.1 backbone. EVs were harvested as described above, and the HS-EV and UC-EVs fractions were combined. EVs were diluted in fresh, complete DMEM and added to a tissue-cultured treated 48-well plate in 100 μL total volume per well. SynTF reporter cells were harvested by washing once in PBS, a short incubation in trypsin (<1 min), and quenching with DMEM. Cells were counted and diluted in DMEM to $2 \times 10^5$ cells per mL; 100 μL of this mixture was added to wells containing 100 μL EVs such that final cell count per well was $2 \times 10^4$. Cell only conditions received 100 μL DMEM instead of EVs. Within each experiment, equal numbers of EVs as determined by NTA were added to each well, typical doses were between $3-4 \times 10^{10}$ EVs per well. Plates were swirled to encourage mixing and were returned to the incubator to culture for 2 d. In parallel, HEK293FTs were transfected with appropriate color compensation controls for flow cytometry as described above ("Transient transfection"). After 2 d, EV-treated cells were imaged on a Keyence BZ-x800 microscope. BZ Series Application software v01.01.00.17 was used, and cells were imaged with either a PlanApo 4 x (NA 0.2), PlanApo 10x (0.45), or PlanFluor 20 x (NA 0.45) objective. Fluorescence images were captured using a dsRed filter cube (Chroma 49005-UF1). Cells were then harvested for flow cytometry. Briefly, cells were harvested with trypsin, quenched with phenol-red free DMEM, and diluted in at least five volumes of fluorescence-activated cell sorting (FACS) buffer (PBS pH 7.4, 2 mM EDTA, 0.05% bovine serum albumin). Cells were pelleted at 150 g for 5 min at 4 °C, the supernatant was decanted, and the samples were stored at 4 °C until flow cytometry analysis.

### Transfected synTF delivery experiments

In a tissue culture-treated 24-well plate, $1.5 \times 10^5$ HEK293FTs were plated in 0.5 mL DMEM and allowed to adhere for 5–8 h. Cells were transfected as described above ("Transient transfection") with equal copy numbers of each synTF (~50 ng), a blue color control, and empty backbone pcDNA plasmid such that each well received 500 ng total DNA. The media was changed the next morning, and cells were harvested for flow cytometry the following morning (~2 days after transfection). Cells were washed in PBS, harvested with trypsin, quenched with phenol red-free DMEM, and diluted in at least five volumes of FACS buffer. Cells were pelleted at 150 g for 5 min at 4 °C, the supernatant was decanted, and the samples were stored at 4 °C until flow cytometry analysis.

### Flow cytometry

Flow cytometry was performed using a BD LSR Fortessa Special Order Research Product (Robert H. Lurie Cancer Center Flow Cytometry Core) running FACSDiva v9.1. Lasers and filter sets used are in Supplementary Table 8. Samples were analyzed using FlowJo V10.8.1 software. For experiments where transcription factor plasmids were directly transfected into reporter cells, ~3000–6000 single cells were analyzed. For all other EV-mediated transcription factor delivery experiments, approximately 30,000–60,000 single cells were analyzed. As shown in Supplementary Fig. 20, HEK293FT cells were first gated for cells using SSC-A versus FSC-A, then gated for singlets using FSC-H versus FSC-A. Subsequent fluorescence gating was used to analyze specific cell populations as needed; for example, gating on miRFP720+ cells to analyze only synTF reporter cells. In all experiments, fluorescence data were compensated for spectral bleedthrough using HEK293FT cells transfected with single color controls. To convert relative units of fluorescence to absolute fluorescence units (i.e., mean molecules of equivalent phycoerythrin-texas red (PE-TR) (MEPTRs) for synTF delivery experiments), Spherotech 9-peak rainbow beads (URCP-100-2H) were diluted ~20x in PBS and run in parallel for each experiment. Calibration curves were generated using the fluorescence intensities from the flow cytometer and the number of fluorophores per bead population as provided by the supplier. The calibration curve was then used to convert mean fluorescence intensities of cells to MEPTRs (Supplementary Fig. 20c).

### Data analysis/statistical analysis

Data were analyzed using Graph Pad Prism 9 or Microsoft Excel. Error bars generally represent the standard error of the mean (SEM); exceptions can be found in the relevant figure captions. Statistical tests and Pearson's correlations were performed in Prism 9; details can be found in the relevant figure captions, the Source Data, and Supplementary Table 9.

### Reporting summary

Further information on research design is available in the Nature Portfolio Reporting Summary linked to this article.

## Data availability

Plasmid maps, plasmid descriptions, and plasmids used in each experiment can be found in Supplementary Data 1. Source data are provided with this manuscript. Due to the large size, raw microscopy and flow cytometry data are not included in the source data but are available upon request. Plasmids generated in this study are deposited with Addgene. Bioinformatic analysis was performed using data reported in the SwissProt (https://www.expasy.org/resources/uniprotkb-swiss-prot), Exocarta (http://www.exocarta.org/), and Raft-Prot (https://raftprot.org/) databases. Other data and unique biological materials (e.g., cell lines) are available on request. Source data are provided with this paper.

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

## Acknowledgements

We thank the Kamat and Leonard labs for providing feedback and proofreading the manuscript. We thank Jan Steinkühler for helpful discussions on how to characterize lipid-protein interactions. We thank Patrick Donahue for generating the landing pad destination backbone used in this study. We thank the Tullman-Ercek Lab for the use of their chemiluminescent imager (Azure c600). This work was supported in part by the McCormick Research Catalyst Program at Northwestern University (J.N.L., N.P.K.); National Science Foundation under Grant No. 1844219 (N.P.K., J.N.L.) and Grant No. 2145050 (N.P.K., T.F.G.); J.A.P., T.F.G., and H. I. E. were supported by NSF Graduate Research Fellowships (DGE-1842165). Any opinions, findings, and conclusions or recommendations expressed in this material are those of the authors(s) and do not necessarily reflect the views of the National Science Foundation. J.A.P. gratefully acknowledges support from the Ryan Fellowship, and the International Institute for Nanotechnology at Northwestern University. T.F.G. was supported in part by the Northwestern University Graduate School Cluster in Biotechnology, Systems, and Synthetic Biology, which is affiliated with the Biotechnology Training Program. This work was supported by the Northwestern University—Flow Cytometry Core Facility supported by Cancer Center Support Grant (NCI CA060553). This work made use of the BioCryo facility of Northwestern University's NUANCE Center, which has received support from the SHyNE Resource (NSF ECCS-2025633), the IIN, and Northwestern's MRSEC program (NSF DMR-1720139). This work was supported by the Northwestern University Sanger Sequencing Facility. This work made use of the Keck Biophysics Facility, a shared resource of the Robert H. Lurie Comprehensive Cancer Center of Northwestern University supported in part by the NCI Cancer Center Support Grant #P30 CA060553. This work was also supported by the Keck Biophysics Facility's Azure Sapphire Imager which is funded by NIH grant 1S10OD026963-01 NIH grant. Biological and chemical analysis was performed in the Analytical bioNanoTechnology Core Facility of the Simpson Querrey Institute at Northwestern University. The U.S. Army Research Office, the U.S. Army Medical Research and Materiel Command, and Northwestern University provided funding to develop this facility and ongoing support is being

received from the Soft and Hybrid Nanotechnology Experimental (SHyNE) Resource (NSF ECCS-1542205).

## Author contributions

J.A.P. and T.F.G. conceived the initial project. J.A.P. performed bioinformatic database analysis. T.F.G. and J.A.P. performed the wet lab experiments. H.I.E. developed the synTF reporter cell line. P.L. and D.B. designed the de novo designed proteins. T.F.G., J.A.P., N.P.K., and J.N.L. planned and analyzed experiments. T.F.G., J.A.P., N.P.K., and J.N.L. wrote the manuscript. N.P.K. and J.N.L. supervised the work.

## Competing interests

J.A.P., T.F.G., N.P.K., and J.N.L. are inventors on a United States patent application (assignee: Northwestern University) that includes technologies reported in this article. D.B. and P.L. are inventors on U.S. patents which cover the computational design of multipass transmembrane proteins and transmembrane pores submitted by the University of Washington. The remaining authors declare no competing interests.
