## [Peer Review File · Nature Communications]

Enhancing extracellular vesicle cargo loading and functional delivery by engineering protein-lipid interactionsREVIEWER COMMENTS

Reviewer #1 (Remarks to the Author):

The manuscript presents a rational framework to engineer protein loading into EVs by taking lipid membrane ordering and raft association into consideration. The end result is improved loading of proteins into EVs via lipidation and their delivery to target cells. Overall, the paper presents a well-executed set of studies and has potential to impact the field of EV-based delivery and therapeutics. As the paper acknowledges, previous studies already showed that lipidating proteins improves their loading into EVs. Thus, it will be important to more precisely determine whether raft association (as opposed to general plasma membrane loading) specifically causes EV loading, and if so, which EV biogenesis pathways are involved. The following revisions are suggested.

1. It is notable that most results in the manuscript are presented from 2 independent experiments. The results from at least $n = 3$ experiments need to be included with statistical analysis to make conclusions.

2. The authors employed DiO labeling of GPMVs from transfected HEK cells and measured protein partitioning into ordered lipid regions. These results were then correlated with quantification of proteins per EV and EV partitioning. To establish a stronger link between raft association and EV loading in cells, it will be important to validate some of the findings by perturbing pathways associated with raft association in cells and test their impact on protein loading into EVs. For instance, will temporary cholesterol depletion in cells by drugs (e.g., Filipin III) or caveolin knockdown result in less protein loading into large vs small EVs?

3. The population pelleted at 120,416 g consists of not only exosomes (the term specifically used for vesicles secreted via multivesicular bodies; MVB) but also other small EVs and non-vesicular particles (PMID: 30951670). Thus, it would be too premature to use the term 'exosome' throughout the manuscript. The same applies to the term 'microvesicle'; rather, the term 'large EV' would be more appropriate (PMID: 30637094). Using unfractionated large or small EVs may confound the interpretation of the results that have implications in EV biogenesis and cargo loading.

4. Exosomes and microvesicles are known to be made via distinct biogenesis pathways (MVB fusion and membrane budding, respectively). So, one would expect that membrane lipid/protein compositions will be different between the two. However, 'exosomes' and 'microvesicles' in the manuscript show a similar degree of lipid ordering (Fig. 1c). Yet, the effect of raft localization and plasma membrane localization on protein loading into EVs seems more prominent in 'microvesicles' (Fig. 5h, Fig. 6). Cholesterol (major lipid raft component) is known to be the main

lipid component of small EVs (PMID: 28342835). Together, it is unclear if lipid ordering is a true indicator of protein loading into large vs small EVs.

5. Supporting the point above, Fig. 5c shows that all of the tested proteins are not well partitioned towards the ordered region (below 0.0) despite that different lipids are tested (in contrast to WT LAT in Fig. 3e). So, does this result mean that protein loading into EVs may have less to do with raft association but more to do with plasma membrane localization?

6. While it makes practical sense to combine large and small EV fractions for transcription factor (TF) delivery experiments in Fig. 7, it will be important to compare TF delivery by large EVs vs. small EVs at least with the PM variant in order to better understand which EV biogenesis pathway likely plays a role in lipid-mediated protein loading.

Reviewer #2 (Remarks to the Author):

In terms of originality, I think that the present paper is worthy of publication. It dwells into the idea that a higher raft partitioning and plasma membrane association can be utilized to increase formation of extracellular vesicles, increase protein association with extracellular vesicles (EV) in EV producing cells and increase the efficiency of transferring proteins from EV to target cells. This concept is quite new and highlights the roles of membrane domains in the formation of extracellular vesicles and how proteins can be modified structurally to increase their transfer to possible target cells through the transport by EVs. However some results need clarification and some additional experiments should be done to reinforce their conclusions and results.

Major comments

1) p.5: fig 1c: The authors should compare the order of extracellular vesicles to the order of the plasma membrane. The comparison towards whole cells is less interesting because these contain also very disordered intracellular membranes. Here, the authors want to show that EV are derived from plasma membrane lipid rafts which should be more ordered than the rest of the plasma membrane and especially intracellular membranes. One way of doing this would be doing GP-imaging of C-Laurdan or analysis of GPMVs produced by these cells.

2) Fig. 3f,g ; Fig S8: Fig. 5d,e,f,g: There need to be some statistical analysis here.

3) Fig S7,IV the plasma membrane mask looks somehow if it were inside the cell. If the dye is endocytosed over the timeframe of the experiment, it is no longer an efficient plasma membrane mask.

4) L. 551: cells. "The addition of lipid tags to the transcription factor impaired the ability to induce reporter expression to different extents across our constructs (compared to the non-lipidated "Sol" synTF control)."

Would lipidation according to the data not increase plasma membrane localization and hence inhibit its localization to the nucleus and prevent transcription in the target cells? Can you prove that some of the products effectively travel to the nucleus? Is it not possible to follow the fluorescent protein constructs that are used to determine GPMV raft partitioning in EVs, to observe fusion to endosomal membranes in target cells and hence follow protein delivery and possible partitioning to the nucleus in target cells?

5) L.567: "We treated reporter cells with EVs containing lipidated ZFs and a surface displayed viral glycoprotein (vesicular stomatitis virus G protein, VSV-G) to promote EV uptake and membrane fusion with recipient endosomal membranes "

- Is the viral protein equally expressed in the EV containing different proteins? If not this would obviously modify the fusion efficiency and hence bias the result. A control should be done.

6) L. 668 "We hypothesize that the PPF-tagged synTFs may be more tightly bound to plasma membranes than PM-tagged synTFs53."

- But in figure 5, PM is as clearly bound to the plasma membrane as the PPF. Why would it be tightly bound to the membrane in the EV producing cells but not in the EV receiving cells? Could the authors please elaborate?

7) Would it be possible to co-transfect cells with PM or PPF to increase the formation of microvesicles and hence to increase the delivery of other proteins to target cells?

8) Fig 6a. It is not clear if the displayed pearson correlation coefficients refer to a significant correlation. Please display the significance of the correlation (p-value) in the graph or elsewhere. This is very important since many interpretations derive from this result.

Minor comments

1) L. 263: “Surprisingly, a raft-partitioning value could not be measured for LAT dCore because this protein did not localize to the plasma membrane and accordingly did not end up in GPMV membranes. This result agrees with cellular trafficking data in Fig. 3c, but it does not match similar studies performed using RBL cells^{22,25}. This discrepancy might suggest that membrane-protein interactions are cell-type specific, perhaps due to interaction with other cellular molecules not explicitly evaluated here.”

The data might also suggest that the lipid composition or membrane properties of these cells are different.

2) L.415&416&712 “Myristylation” should be spelled: “myristoylation”

3) Fig. 1b: Please be more precise what “Biophysical evaluation of select proteins” means. Here in this article it is rather determining structural features of protein transmembrane domains.

4) L.183: phosphorylation was chosen as they “affect protein-membrane interactions, cellular trafficking, and lipid raft localization”:

- It is not sure if phosphorylation affects raft partitioning. It was rather discovered as a feature of raft-like proteins and which is possibly enriched in “raft-proteins” because they are signaling domains. (ref 21)

5) Fig 4b,c “protein hydrophobic thickness” should be replaced by “protein transmembrane domain hydrophobic length”

6) L. 387: “This outcome that transmembrane domain length is not always predictive of EV loading is consistent with our bioinformatic data, since transmembrane domain length did not correlate with raft or EV association for multi-pass transmembrane proteins (Fig. 2c).

- This has also been previously observed in (ref 21).

Reviewer #3 (Remarks to the Author):

In this paper, the authors investigated the role of membrane-protein interaction to enhance EV cargo loading through bioinformatics, biophysical evaluation, and functional delivery of synthetic cargo molecules. The high degree of the association of proteins in EVs and lipid rafts supported the idea that modulating lipid raft affinity of proteins affects cellular trafficking and loading into extracellular vesicles. With this, the authors went on to engineer transcriptional factors (TFs) for EV loading by adding lipid tags and showed their efficient EV loading and functional delivery to reporter cells. The empirical determination for the choice of lipid tags seems to be required depending on proteins of interest, while the limitation and the future directions are clearly stated. Most, if not all of their conclusions are supported by data. The methods are comprehensive and the experimental details in the text are almost complete. Overall, this is a well-designed study that brings innovations in engineering EVs and provides deeper insights into EV cargo sorting mechanisms. I would support the publication of this paper in Nature Communications only after the authors fully addressed my concerns.

Major comments

1. The authors should demonstrate the compatibility of their engineered EVs in reporter mice systems or clinically relevant mice models.
2. The mechanism behind the dissociation of lipidated cargo from EVs in recipient cells is not clear. One might argue that the functional cargo transfer is mediated by the non-lipidated fraction (e.g., cargo with lipid tags but not lipidated) in EVs, which may explain why the functional transfer is very inefficient (less than 10%). In this context, the lipidation states of engineered EV cargos were not investigated at all.
3. C-laurdan staining suggests that EVs from HEK293FT cells are liquid ordered (lipid raft-like). This conclusion could be further strengthened by in-depth lipidomic analysis of EVs.
4. The authors should repeat experiments for EV incorporation and EV partitioning, and should perform statistical analysis.
5. About 20% of raft-associated proteins were not associated with exosomes (Figure 1d). Could the authors analyze key features of these proteins and further highlight the importance of lipidation and transmembrane domain length (especially for single-pass TM proteins) as the determinants for proteins to be associated with both rafts and EVs?
6. The claim (line 500) that “this is a generalizable mechanism governing protein loading into EVs” is not justified well, as this hypothesis was tested only for LAT, Halo and synTF.

Minor comments

1. The authors could improve data presentation by showing individual data points.

2. The rationale behind the use of 4TMD and 12 TMD in Figure 4 is missing from the text.
3. Supplementary figures 10 and 12 may be redundant.

Response to Reviewers

Reviewer #1 (Remarks to the Author):

The manuscript presents a rational framework to engineer protein loading into EVs by taking lipid membrane ordering and raft association into consideration. The end result is improved loading of proteins into EVs via lipidation and their delivery to target cells. Overall, the paper presents a well-executed set of studies and has potential to impact the field of EV-based delivery and therapeutics. As the paper acknowledges, previous studies already showed that lipidating proteins improves their loading into EVs. Thus, it will be important to more precisely determine whether raft association (as opposed to general plasma membrane loading) specifically causes EV loading, and if so, which EV biogenesis pathways are involved. The following revisions are suggested.

Thank you for your thoughtful review and for highlighting the contribution of our study on the EV-delivery field, particularly in the context of existing literature. Based upon your suggestions, we have performed additional wet lab experiments, data analysis, and added discussion to this revised manuscript.

1. It is notable that most results in the manuscript are presented from 2 independent experiments. The results from at least $n = 3$ experiments need to be included with statistical analysis to make conclusions.

We agree with the reviewer's central point that conclusions should be justified by statistical support; to address this comment, we updated the manuscript to include the results of statistical analyses in key places including figures Fig. 1c, d; Fig. 3f, g; Fig 4d, e; Fig. 5d, e, f, g; Fig. 7c, Fig. S4-10. However, we respectfully disagree that an n of 3 independent experiments is required to draw conclusions. We have carefully designed the experiments presented in the study to capture the error within each type of experiment and across experimental replicates, and our conclusions are contextualized within that measured variation. While it is possible that performing additional experiments would enable us to draw additional conclusions about other phenomena (i.e., to evaluate whether minor trends are meaningful or simply variation), we limited our conclusions to conservative statements which are already supported by statistically significant comparisons in the data as presented.

2. The authors employed DiO labeling of GPMVs from transfected HEK cells and measured protein partitioning into ordered lipid regions. These results were then correlated with quantification of proteins per EV and EV partitioning. To establish a stronger link between raft association and EV loading in cells, it will be important to validate some of the findings by perturbing pathways associated with raft association in cells and test their impact on protein loading into EVs. For instance, will temporary cholesterol depletion in cells by drugs (e.g., Filipin III) or caveolin knockdown result in less protein loading into large vs small EVs?

Thank you for this interesting suggestion. To address this comment, we performed an experiment in which Sol, M, or PM HaloTag variants of our probe protein were expressed via transient transfection in HEK293FTs. Cells were cultured with Filipin III for approximately 30 hours, and then harvested cell lysates and EVs were analyzed for HaloTag expression and loading, respectively, via fluorescence analysis of protein gels as we did in other experiments. Unfortunately, we observed that treating transfected HEK293FTs with Filipin III was highly toxic even at a low dose of 1 μ M (Response Fig. 1a). When this experiment was run under conditions of 0.5 μ M Filipin III, we did not observe meaningful differences in

HaloTag loading for Sol, M, or PM variants when comparing Filipin III-treated or vehicle-treated conditions (Response Fig. 1b). This result is not entirely conclusive, however, given that typical lipid raft disruption studies in HEK293s use Filipin III at much higher doses (10 μ M) and for a short period of time¹, we surmise that the experimental conditions used to evaluate EV loading (including transfection method, cell type, culture time, etc.) are not compatible with the use of Filipin III to address this question. For example, it is possible that the highest tolerable dose (0.5 μ M Filipin III) is not sufficient to disrupt lipid rafts in a manner that enables us to perturb raft-associated HaloTag loading due to the time scale required to generate EVs.

Although we could not use Filipin III as a tool to further evaluate the connection between lipidation and EV loading in our system, our conclusions are plausible based upon other similar studies probing the impact of lipidation in distinct systems. For example, Ye et al. investigated how Filipin III treatment modulated loading of two peripheral proteins into EVs isolated from PC-3 cells²; we note that this experiment probed natively expressed proteins (not introduced by transfection). They found that Filipin III treatment disrupted EV-loading of Fyn kinase, which possesses a myristic acid and 2 palmitic acids on its N-terminus, but this disruption was not observed for Src kinase, which possess a single N-terminal myristic acid (the same terminus as our M tag). Based on these results, one would hypothesize that Fyn is raft associated, while Src kinase is not. Our observations support that conclusion; we evaluated the Src kinase tag and observed that it does not mediate association with lipid rafts. In other supporting evidence, the N-terminus of Fyn kinase has been shown to drive raft association in GPMVs³. Overall, our results are concordant with these two additional studies demonstrating that raft associated protein loading into EVs is disrupted by Filipin III, while non-raft associated protein loading is not.

Response Figure 1. Filipin iii treatment is highly toxic to transfected HEK293FTs and does not cause meaningful differences in HaloTag loading for Sol, M, or PM variants at a tolerated dose. a Micrographs of HEK293FTs transfected with plasmids encoding the PM HaloTag variant, followed by treatment with 0.5 pM or 1.0 pM Filipin III (F) to disrupt lipid rafts or a vehicle (V), DMSO, control. Treatment occurred for approximately 30 h before harvesting cell lysates and EVs. 1 pM Filipin III demonstrates toxicity to transfected HEK293FTs. Scale bar is 500 pm. **b** Fluorescent protein gels for cell lysates (left, 5 pg protein loaded) and EVs (right, 5×10^8 EVs loaded) for Sol, M, and PM-tagged HaloTag. When producer cells were treated with 0.5 pM Filipin III, there was minimal effect on HaloTag expression in the producer cell lysate or loading into EVs.

3. The population pelleted at 120,416 g consists of not only exosomes (the term specifically used for vesicles secreted via multivesicular bodies; MVB) but also other small EVs and non-vesicular particles (PMID: 30951670). Thus, it would be too premature to use the term ‘exosome’ throughout the manuscript. The same applies to the term ‘microvesicle’; rather, the term ‘large EV’ would be more appropriate (PMID: 30637094). Using unfractionated large or small EVs may confound the interpretation of the results that have implications in EV biogenesis and cargo loading.

Thank you for the comment. We agree that based on our present characterization, it is imprecise to use the terms ‘exosome’ and ‘microvesicle.’ We have altered the text to refer to vesicle populations as “high-speed centrifugation EV fraction” (HS-EV, formerly microvesicle) and “ultracentrifugation EV fraction” (UC-EV, formerly exosome). By referring to vesicle populations by the way each was purified, we are more accurately describing by physical features (density), rather than implying from where in the cell each population is derived. We use these terms rather than size references because in this system HS-EVs and UC-EVs are similar in size but differ in density (as this is what vesicles purifications were based on).

4. Exosomes and microvesicles are known to be made via distinct biogenesis pathways (MVB fusion and membrane budding, respectively). So, one would expect that membrane lipid/protein compositions will be different between the two. However, ‘exosomes’ and ‘microvesicles’ in the manuscript show a similar degree of lipid ordering (Fig. 1c). Yet, the effect of raft localization and plasma membrane localization on protein loading into EVs seems more prominent in ‘microvesicles’ (Fig. 5h, Fig. 6). Cholesterol (major lipid raft component) is known to be the main lipid component of small EVs (PMID: 28342835). Together, it is unclear if lipid ordering is a true indicator of protein loading into large vs small EVs.

First off, we appreciate and generally agree with all points made in this comment as they pertain to expectations based upon prior literature. Generalized polarization of Laurdan labeled membranes is a classical measurement to determine lipid order. Lipid order is not equivalent to lipid composition. We thus do not endeavor to equate the *compositions* of both vesicle populations, but rather we found that both vesicle populations are ordered (which is often described as lipid-raft like). For example, a membrane could be highly ordered, and possess a high Laurdan GP, if it is composed of long, fully saturated lipids. A separate membrane could have a similar Laurdan GP with more unsaturated fatty acid chains but higher cholesterol content. Here in Figure 1c, we are trying to demonstrate that both vesicle populations are more ordered and lipid-raft like, compared to other cellular membranes. This can all be true while acknowledging that the biogenesis, lipid, and protein composition of each vesicle fraction likely differs and contributes to differences in loading.

We have added the following sentence to clarify this point:

“Both vesicle fractions were found to have similar Laurdan GP values, despite the fact that their biogenesis, lipid composition, and protein content may likely differ⁴⁻⁸.”

5. Supporting the point above, Fig. 5c shows that all of the tested proteins are not well partitioned towards the ordered region (below 0.0) despite that different lipids are tested (in contrast to WT LAT in Fig. 3e). So, does this result mean that protein loading into EVs may have less to do with raft association but more to do with plasma membrane localization?

We agree that our data demonstrate that protein loading into EVs is strongly related to protein localization to the plasma membrane. The data in Fig. 5c demonstrate that the PM and PPF HaloTag constructs prefer the L_o phase more than the M and GG Halo constructs. The correlation for EV loading with plasma membrane localization and raft association can be further seen in Supplementary Figure 18. Fig. S18 demonstrates that protein loading more strongly correlates with plasma membrane localization than lipid raft partitioning. This is not necessarily contradictory, as trafficking to the plasma membrane and raft localization are related. Proteins which associate with lipid rafts more are more effectively trafficked to the plasma membrane⁹.

DTT was used in the process employed to generate GPMVs, and DTT can cleave disulfides and thioesters leading to depalmitoylation of proteins¹⁰. This possibility could confer an underestimate of raft association for palmitoylated proteins (LAT, PM, and PPF) in our GPMV analysis. Thus, such proteins could be more strongly raft-associated in the cellular context, and raft-association could be more correlated to EV loading than we observed in this GPMV assay.

We have added the following text and citation to clarify this point:

Protein association with lipid rafts has been shown to enhance plasma membrane localization⁹. Because of this relationship, it is unsurprising that both raft and plasma membrane localization correlate with protein loading into EVs and this observation supports the hypothesis that lipid rafts play a role in EV biogenesis⁸.

Furthermore, the raft association of palmitoylated proteins is potentially underestimated; DTT was used in the process to form GPMVs, and DTT can cleave disulfides and thioesters leading to depalmitoylation of proteins¹⁰. Thus, palmitoylated proteins (LAT, PM, and PPF) are potentially more strongly raft associated in cellular systems and more correlated to EV loading than was observed in this assay.

6. While it makes practical sense to combine large and small EV fractions for transcription factor (TF) delivery experiments in Fig. 7, it will be important to compare TF delivery by large EVs vs. small EVs at least with the PM variant in order to better understand which EV biogenesis pathway likely plays a role in lipid-mediated protein loading.

We agree that it would be interesting to compare loading and the ability of each vesicle population to deliver cargo. Here, we focused on understanding how different lipid-protein interactions could be modulated to enhance loading, and thus our conclusions were written so as not to depend upon that further analysis. We are planning future studies to better understand how choice of EV subpopulation affects functional delivery in different contexts, which include different methods of EV fractionation (i.e., to better dissect the relationships with routes of biogenesis), different cell sources, and different culture conditions, each of which likely affects the phenomena introduced in this first study.

Reviewer #2 (Remarks to the Author):

In terms of originality, I think that the present paper is worthy of publication. It dwells into the idea that a higher raft partitioning and plasma membrane association can be utilized to increase formation of extracellular vesicles, increase protein association with extracellular vesicles (EV) in EV producing cells and increase the efficiency of transferring proteins from EV to target cells. This concept is quite new and highlights the roles of membrane domains in the formation of extracellular vesicles and how proteins can be modified structurally to increase their transfer to possible target cells through the transport by EVs. However some results need clarification and some additional experiments should be done to reinforce their conclusions and results.

Thank you for your supportive comments regarding the novelty and potential impact of this study.

Major comments

1) p.5: fig 1c: The authors should compare the order of extracellular vesicles to the order of the plasma membrane. The comparison towards whole cells is less interesting because these contain also very disordered intracellular membranes. Here, the authors want to show that EV are derived from plasma membrane lipid rafts which should be more ordered than the rest of the plasma membrane and especially intracellular membranes. One way of doing this would be doing GP-imaging of C-Laurdan or analysis of GPMVs produced by these cells.

Thank you for this suggestion. To address this comment, we have added C-Laurdan analysis of GPMVs to Figure 1c. GPMVs are significantly more ordered than are whole cells, as expected ($p=0.027$). However, both EV populations are more ordered than GPMVs, which supports the idea that EVs are more ordered than plasma membrane-derived membrane-mimetic such as GPMVs. This is likely because GPMVs possess both ordered and disordered lipid domains.

New Figure 1:

Figure 1. EV membrane physical properties and protein content mirror those of lipid rafts. a Schematic of our hypothesis that lipid-raft association could be used as a handle to load proteins into EVs. **b** Overall analysis and experimental workflow. We used RaftProt 2.0 and Exocarta to understand features of proteins found in EVs and lipid rafts (I); we then built a library of structurally diverse proteins to understand how such features affect protein trafficking, interactions with lipid rafts, and loading into EVs (II); applying these design rules, we demonstrate how lipid-protein interactions can be used to functionally deliver cargo to cells via EVs (III). **c** Laurdan generalized polarization (GP) for ordered liposomes (L_o) composed of 70 mol% DPPC/30 mol% Chol, disordered liposomes (L_D) composed of 70 mol% DOPC/30 mol% Chol, HEK293FT cells, giant plasma membrane vesicles (GPMVs) and vesicles from the high-speed centrifugation EV fraction (HS-EVs) and ultracentrifugation EV fraction (UC-EVs) derived from HEK293FT cells. HS-EV and UC-EVs had high laurdan GP, similar to L_o membranes. Each dot represents an independent experiment. A one-way ANOVA was performed to compare the Laurdan GP of HS-EVs and UC-EVs to the Laurdan GP of all other membranes measured, and comparisons were evaluated using the Sidak multiple comparisons correction. Both EV populations were significantly different from all other vesicle populations (****, $p < 0.0001$). **d** The frequency at which human membrane-associated proteins and raft proteins are found in EVs as calculated via bioinformatic analysis. Raft associated proteins are more frequently found in EVs compared to a random selection of human membrane-associated proteins. Each dot represents a separate query of 100 human proteins. An unpaired parametric t-test was performed to compare the fraction of all proteins and raft associated proteins found in exosomes. The fraction of raft associated proteins found in exosomes was significantly

different than the fraction of all proteins ($p < 0.0001$). $n \geq 3$, error bars represent the standard error of the mean (SEM) throughout the figure.

2) Fig. 3f,g ; Fig S8: Fig. 5d,e,f,g: There need to be some statistical analysis here. We have added statistical analysis for these figures, as well as Fig. 1c, d; Fig 4d, e; Fig. 7c, Fig. S4-10.

3) Fig S7,IV the plasma membrane mask looks somehow if it were inside the cell. If the dye is endocytosed over the timeframe of the experiment, it is no longer an efficient plasma membrane mask.

We have readjusted these images to better represent the plasma membrane stain. We do not have evidence that the dye is endocytosed over the course of the experiment. Furthermore, we would like to highlight that our staining and experimental methodology is validated by Figure 3c, as we were able to recapitulate previously reported analyses of the plasma membrane localization results of LAT proteins (Fig. 3c)¹¹.

4) L. 551: cells. “The addition of lipid tags to the transcription factor impaired the ability to induce reporter expression to different extents across our constructs (compared to the non-lipidated “Sol” synTF control).”

Would lipidation according to the data not increase plasma membrane localization and hence inhibit its localization to the nucleus and prevent transcription in the target cells? Can you prove that some of the products effectively travel to the nucleus? Is it not possible to follow the fluorescent protein constructs that are used to determine GPMV raft partitioning in EVs, to observe fusion to endosomal membranes in target cells and hence follow protein delivery and possible partitioning to the nucleus in target cells?

We agree that lipidation likely inhibits localization of constructs to the nucleus in both the direct transfection and delivery experiments (Fig. 7b and Fig. 7f respectively). While line 551 refers to experiments where plasmid DNA encoding the synTFs were transfected into reporter cells, we are interpreting the reviewer’s questions to be referencing delivery of synTFs to reporter cells via EVs. We hypothesize that proteins which are loaded into EVs are lipidated. However, once in an EV or delivered to a cell, we are not yet certain if or how the lipidation state of each construct changes. We found that the PM tag yielded the highest functional delivery, but PPF was found to load best into EVs. It is very possible that the PPF tagged transcription factors are also delivered best to receiver cells but remain tethered to the membrane, and thus do not initiate gene transcription. We comment on this further in response to your major comment 6 below.

In order to initiate transcription of the downstream dsRedExp2 protein, the transcription factors delivered by EVs must travel to the nucleus. Thus, this assay confirms that the protein cargo is able to travel to the nucleus. Tracking the delivery of a fluorescent protein is an interesting suggestion that could provide insights beyond the confirmation included here, and we are currently planning to pursue this in a follow up study (obtaining the signal-to-noise required for visualization is substantial work beyond the scope of this study).

To capture these ideas, we have added the following text to clarify the above points:

The differences in functionality between lipidated synTFs may be partially attributed to lipidation-mediated sequestration of synTFs to a membrane, preventing them from entering the nucleus.

5) L.567: “We treated reporter cells with EVs containing lipidated ZFs and a surface displayed viral glycoprotein (vesicular stomatitis virus G protein, VSV-G) to promote EV uptake and membrane fusion with recipient endosomal membranes ”

- Is the viral protein equally expressed in the EV containing different proteins? If not this would obviously modify the fusion efficiency and hence bias the result. A control should be done.

Thank you for this suggestion. To answer this question, we performed a new experiment in which we probed each of our EV populations for VSV-G loading via a western blot (n = 3). The images are available in Supplementary Figure 23a and have been quantified in panel b of the same figure. Interestingly, we see slight differences in VSV-G loading into EVs depending on the lipidation-tagged SynTF variant that was co-transfected, despite transfecting in equal amounts of VSV-G-encoding plasmid into the EV producer cells across all conditions. We are unaware of evidence in prior literature demonstrating that small differences in VSV-G loading (as observed here) drive substantial differences in functional delivery. Importantly, these experiments demonstrated that VSV-G loading is similar for many of the synTF variants, and so differences in functional delivery are unlikely to be due to differences in VSV-G loading (e.g., VSV-G loading is not likely to account for differences between PM and PPF delivery).

We have added the following text to the manuscript to contextualize these additional experiments:

Small differences in VSV-G loading into EVs were observed across the SynTF variants (Supplementary Fig. 23); these minor variations are unlikely to explain the large differences in functional delivery observed in Fig. 7f.

6) L. 668 “We hypothesize that the PPF-tagged synTFs may be more tightly bound to plasma membranes than PM-tagged synTFs⁵³.”

- But in figure 5, PM is as clearly bound to the plasma membrane as the PPF. Why would it be tightly bound to the membrane in the EV producing cells but not in the EV receiving cells? Could the authors please elaborate?

We agree that Figure 5b demonstrates that PM and PPF both localize to the plasma membrane to a similar extent in the producer cell. Notably, these data represent a quasi-equilibrium state when the cargo protein is in excess, and the driving forces for association/dissociation would be very different in recipient cells when the cargo protein is at a low (zero, initially) concentration. We hypothesize that PM may also be more likely to dissociate from membranes because: (i) PM is modified with two lipids (1 myristic and 1 palmitic acid) compared to three lipids for PPF (2 palmitic acids and 1 farnesyl), and (ii) palmitoylation is a reversible post-translational modification¹². We have added the following text to clarify this point.

We hypothesize that the PPF-tagged synTFs may be more tightly bound to membranes than PM-tagged synTFs in reporter cells because the PPF tag is composed of 3 lipids (2 palmitic acids and 1 farnesyl) compared to two for PM (1 myristic and 1 palmitic acid)¹³.

7) Would it be possible to co-transfect cells with PM or PPF to increase the formation of microvesicles and hence to increase the delivery of other proteins to target cells?

Thank you for this question. We hypothesize that this is possible and are currently thinking about ways to leverage this phenomenon. We speculate this possibility in the following passage of text:

Microvesicles, or EVs which are derived from the plasma membrane, are likely enriched within the HS-EV fraction. The observed increase in HS-EVs could be explained by enhanced microvesicle formation due to increased membrane budding as a result of protein crowding on the plasma membrane, and this observation suggests a possible strategy to intentionally increase EV production¹⁴.

8) Fig 6a. It is not clear if the displayed pearson correlation coefficients refer to a significant correlation. Please display the significance of the correlation (p-value) in the graph or elsewhere. This is very important since many interpretations derive from this result.

Thank you for raising this excellent point. We have added an additional supplementary figure (Fig. S19) to present the p-values of these correlations. Indeed, not all Pearson's correlation coefficients refer to a significant correlation. However, key correlations, such as the correlation between protein partition and plasma membrane localization and the correlation between protein loading/partition and raft localization, are significant. We do not believe that these values should be necessarily linearly correlated, yet Pearson's correlation provides a simplistic model to demonstrate that some of these values are indeed significantly related.

a. Figure 6a- all proteins combined								
	Lysate Expression	PM Localization	ER Localization	Raft Localization	EV number - HS-EV	EV Number - UC-EV	EV Size - HS-EV	EV Size - UC-EV
HS-EV loading	0.6221	0.1908	0.7386	0.0040	0.5096	0.0623	0.9959	0.9239
UC-EV Loading	0.3747	0.2557	0.6723	0.0100	0.3973	0.0114	0.9769	0.5776
HS-EV Partition	0.0004	0.0158	0.7546	0.0130	0.0003	0.2205	0.0021	0.0192
UC-EV Partition	0.0079	0.0242	0.7253	0.0148	0.0066	0.0326	0.0111	0.0348
b. LAT proteins								
	Lysate Expression	PM Localization	ER Localization	Raft Localization	EV number - HS-EV	EV Number - UC-EV	EV Size - HS-EV	EV Size - UC-EV
HS-EV loading	0.2578	0.0993	0.7595	0.0548	0.0425	0.7033	0.0037	0.0876
UC-EV Loading	0.2881	0.1653	0.5476	0.2468	0.0150	0.7884	0.0277	0.0820
HS-EV Partition	0.5779	0.0852	0.1810	0.0122	0.6145	0.5225	0.1089	0.6491
UC-EV Partition	0.9461	0.0828	0.9024	0.1136	0.1877	0.7667	0.0995	0.3802
c. De novo designed proteins								
	Lysate Expression	PM Localization	ER Localization	Raft Localization	EV number - HS-EV	EV Number - UC-EV	EV Size - HS-EV	EV Size - UC-EV
HS-EV loading	0.2510	0.1199	0.0038	0.2531	0.1724	0.9145	0.7800	0.0363
UC-EV Loading	0.7478	0.4089	0.0192	0.9528	0.0743	0.7775	0.9456	0.1905
HS-EV Partition	0.1934	0.1471	0.0050	0.2867	0.1789	0.9392	0.7773	0.0398
UC-EV Partition	0.3618	0.4294	0.0097	0.8150	0.1067	0.9244	0.9398	0.0945
d. Lipidated peripheral proteins								
	Lysate Expression	PM Localization	ER Localization	Raft Localization	EV number - HS-EV	EV Number - UC-EV	EV Size - HS-EV	EV Size - UC-EV
HS-EV loading	0.0836	0.0126	0.5396	0.0902	0.0015	0.2312	0.0001	0.0274
UC-EV Loading	0.0079	0.0034	0.4387	0.0365	0.0001	0.6730	0.0023	0.1646
HS-EV Partition	0.0002	0.0086	0.3765	0.0323	0.0021	0.5949	0.0871	0.7554
UC-EV Partition	0.0025	0.0532	0.4493	0.0785	0.0336	0.2947	0.3120	0.8575

Supplementary Figure 19. P-values for correlation data presented in Figure 6a and Supplementary Figure 18. P-values for Pearson's correlation in a Figure 6a for all proteins analyzed in this study, b LAT proteins (LAT WT, LAT C26A, LAT dCore, LAT High ASA), c *de novo* designed proteins (24 Å TMD4, 32 Å TMD4, 40 Å TMD4, 24 Å TMD12, 40 Å TMD12), and d lipid tagged HaloTag proteins (Sol, M, PM, G, PPF).

Minor comments

1) L. 263: "Surprisingly, a raft-partitioning value could not be measured for LAT dCore because this protein did not localize to the plasma membrane and accordingly did not end up in GPMV membranes. This result agrees with cellular trafficking data in Fig. 3c, but it does not match similar studies performed using RBL cells^{22,25}. This discrepancy might suggest that membrane-protein

interactions are cell-type specific, perhaps due to interaction with other cellular molecules not explicitly evaluated here.”

The data might also suggest that the lipid composition or membrane properties of these cells are different.

Thank you for raising this point. This is what we were trying to suggest, but the wording was unclear. We have reworded this sentence as follows:

This discrepancy might suggest that membrane-protein interactions are cell-type specific, perhaps due to differences in membrane composition and membrane physiochemical properties between cell types.

2) L.415&416&712 “Myristylation” should be spelled: “myristoylation”

Thank you for pointing this out. We have corrected those misspellings.

3) Fig. 1b: Please be more precise what “Biophysical evaluation of select proteins” means. Here in this article it is rather determining structural features of protein transmembrane domains.

We have edited Figure 1b to now read “Evaluation of membrane protein structural features” to be more specific. The new Figure 1 is below.

Figure 1. EV membrane physical properties and protein content mirror those of lipid rafts.

4) L.183: phosphorylation was chosen as they “affect protein-membrane interactions, cellular trafficking, and lipid raft localization”:
 - It is not sure if phosphorylation affects raft partitioning. It was rather discovered as a feature of raft-like proteins and which is possibly enriched in “raft-proteins” because they are signaling domains. (ref 21)

Thank you for pointing this out. We have rephrased the sentence to make it more accurate.

These properties were chosen as they affect protein-membrane interactions, cellular trafficking, lipid raft localization, or have been identified as a feature of lipid-raft like proteins^{3,9,11,15–18}.

5) Fig 4b,c “protein hydrophobic thickness” should be replaced by “protein transmembrane domain hydrophobic length”

Thank you for this suggestion. We have changed the graph labels to be “Protein transmembrane domain hydrophobic thickness (Å).”

6) L. 387: “This outcome that transmembrane domain length is not always predictive of EV loading is consistent with our bioinformatic data, since transmembrane domain length did not correlate with raft or EV association for multi-pass transmembrane proteins (Fig. 2c).
- This has also been previously observed in (ref 21).

Thank you for identifying this additional context. We have added the following sentence to better contextualize our finding with previous work done in ref 21.

Furthermore, this trend concords with previous observations by Yurtsever and Lorent¹⁵.

Reviewer #3 (Remarks to the Author):

In this paper, the authors investigated the role of membrane-protein interaction to enhance EV cargo loading through bioinformatics, biophysical evaluation, and functional delivery of synthetic cargo molecules. The high degree of the association of proteins in EVs and lipid rafts supported the idea that modulating lipid raft affinity of proteins affects cellular trafficking and loading into extracellular vesicles. With this, the authors went on to engineer transcriptional factors (TFs) for EV loading by adding lipid tags and showed their efficient EV loading and functional delivery to reporter cells. The empirical determination for the choice of lipid tags seems to be required depending on proteins of interest, while the limitation and the future directions are clearly stated. Most, if not all of their conclusions are supported by data. The methods are comprehensive and the experimental details in the text are almost complete. Overall, this is a well-designed study that brings innovations in engineering EVs and provides deeper insights into EV cargo sorting mechanisms. I would support the publication of this paper in Nature Communications only after the authors fully addressed my concerns.

Thank you for your thoughtful consideration of how our study contributes to the field's understanding of native and engineered EV cargo sorting.

Major comments

1. The authors should demonstrate the compatibility of their engineered EVs in reporter mice systems or clinically relevant mice models.

We agree that exploring the translational potential for this technology will ultimately require preclinical evaluations in relevant animal models. In this work, we focused on characterizing the underlying biophysical interactions of protein loading to better understand how to load and deliver EV encapsulated cargo *in vitro* (rather than optimizing delivery of a specific translationally motivated cargo for a specific application). We are planning to extend this work and evaluate *in vivo* delivery of therapeutic cargos loaded using the concepts elucidated in this study, but given the substantial amount of technical work required to execute such experiments in a meaningful manner (e.g., to quantitatively evaluate the extent to which lipidation improves performance over a baseline condition) we would respectfully argue that such work is outside the scope of the current, foundational study.

2. The mechanism behind the dissociation of lipidated cargo from EVs in recipient cells is not clear. One might argue that the functional cargo transfer is mediated by the non-lipidated fraction (e.g., cargo with lipid tags but not lipidated) in EVs, which may explain why the functional transfer is very inefficient (less than 10%). In this context, the lipidation states of engineered EV cargos were not investigated at all.

We agree that it would be interesting to explore the lipidation states of engineered EV cargos in the future. In this study, we found that lipidation of peripheral proteins has a clear impact on protein loading, and that the presence of a lipidation tag influences the degree of functional delivery in our model system. We do not yet know if proteins remain lipidated in the EV or after delivery to a recipient cell, and that is an important question that we plan to explore in a follow-on study.

With the caveat that a direct comparison to other systems is not feasible due to differences in assay setup (e.g., genomic vs. transient reporter, choice of membrane fusion protein, luciferase expression instead of fluorescent protein expression as an output, different definitions of EV number, etc.), we would nonetheless note that while PM's degree of activation in Fig 7f (~7.4%) may seem low, its fold induction compared to the mock condition is quite high (54-fold). Other groups have reported fold induction for EV-mediated delivery of synthetic transcription factors in HEK293 cells. For a non-enzymatic (i.e., non signal-amplifying) output that is most similar to the one discussed in this manuscript, a fold induction of 3.2 has been reported¹⁹. Other groups using luciferase outputs (i.e., signal-amplifying) with stronger activation domains have reported a fold induction of 14 to 160^{20,21}. Thus, the lipidated synTFs in our study have a comparable or higher ON/OFF signal than the most similar other systems, and they have a competitive fold induction when compared to signal-amplifying systems.

To address this comment and capture these considerations, we have added the following text to both provide speculative explanations and delineate what we do not yet know about the lipidation state of the proteins:

The differences in functionality between lipidated synTFs may be partially attributed to lipidation-mediated sequestration of synTFs to a membrane, preventing them from entering the nucleus.

We hypothesize that the PPF-tagged synTFs may be more tightly bound to membranes than PM-tagged synTFs in reporter cells because the PPF tag is composed of 3 lipids (2 palmitic acids and 1 farnesyl) compared to two for PM (1 myristic and 1 palmitic acid)¹³.

3. C-laurdan staining suggests that EVs from HEK293FT cells are liquid ordered (lipid raft-like). This conclusion could be further strengthened by in-depth lipidomic analysis of EVs.

We agree that it would be interesting to perform in-depth lipidomic analysis on our EVs, but we would respectfully argue that such analysis is out of the scope of the present work. Lipidomic analysis of EVs has previously been reported and demonstrates that the EVs are composed largely of cholesterol and saturated lipids^{22,23}. Laurdan allows for the measurement of bulk lipid properties and has been used widely in the membrane biophysics community to characterize lipid rafts. We believe that the characterization of our vesicles with C-Laurdan, coupled with published lipidomics analysis of EVs, allows us to strongly conclude that EV membranes are ordered without performing further lipidomic analysis.

4. The authors should repeat experiments for EV incorporation and EV partitioning, and should perform statistical analysis.

Thank you for the suggestion to perform additional statistical testing on the datasets provided. We have therefore added additional statistical analysis to Fig. 1c,d; Fig. 3f,g ; Fig 4d,e, Fig. 5d,e,f,g; and Fig S4-10.

Overall, we carefully designed the experiments presented in the study to capture the error within each type of experiment and across experimental replicates, and our conclusions are contextualized within that

measured variation. While it is possible that performing additional experiments would enable us to draw additional conclusions about other phenomena (i.e., to evaluate whether minor trends are meaningful or simply variation), we limited our conclusions to conservative statements which are already supported by statistically significant comparisons in the data as presented.

5. About 20% of raft-associated proteins were not associated with exosomes (Figure 1d). Could the authors analyze key features of these proteins and further highlight the importance of lipidation and transmembrane domain length (especially for single-pass TM proteins) as the determinants for proteins to be associated with both rafts and EVs?

Thank you for this comment. We have added this additional analysis to the supplement (Supplementary Figures 8-11). We find that transmembrane proteins associated with lipid rafts found in EVs are not significantly different from those not found in EVs (Note: there are significant comparisons present, but not when comparing plasma membrane (PM) or internal membrane (IM) associated proteins between groups). This may suggest that the raft associated proteins which are excluded from EVs are not excluded on the basis of biophysical properties or mechanisms evaluated in this study (Fig. S8, 9). However, peripheral membrane proteins associated with lipid rafts which were found in EVs were significantly more myristoylated and prenylated than were raft proteins not found in EVs (Fig. S10). We also evaluated the percentage of proteins within each group that were associated with the plasma membrane. We found that a smaller percentage of raft associated, non-EV proteins were found in the plasma membrane compared to raft associated proteins that have been identified in EVs (Fig S11). This further underscores the importance of plasma membrane localization for protein loading into EVs. It has previously been shown that membrane raft association is a determinant of plasma membrane localization⁹. In this light, one could hypothesize that there exists a gradient of raft association, and weakly raft associated proteins may not load into EVs.

6. The claim (line 500) that “this is a generalizable mechanism governing protein loading into EVs” is not justified well, as this hypothesis was tested only for LAT, Halo and synTF.

We have adjusted the manuscript to reflect the scope of the study more accurately. The claim now reads as follows:

Furthermore, the strong agreement between our bioinformatic analysis (Fig. 1d, Fig. 2) and EV loading data for a diverse set of natural and *de novo* designed proteins (Fig. 3, 4, 5) suggests that strategies presented here may be applied to improve loading of a wide array of protein classes into EVs.

Minor comments

1. The authors could improve data presentation by showing individual data points.

Thank you for this comment. We have added an additional supplementary figure (Supplementary Figure 4) presenting the data in Figure 2 as individual points. We believe that the amount of data for these plots makes it more difficult to interpret and thus have kept the original figure in the main text. All other main text figures present data as individual points when possible.

Below please find the new Supplementary Figure 4:

Single-pass transmembrane proteins

Multi-pass transmembrane proteins

Peripheral membrane proteins

Supplementary Figure 4. Plots of protein structural features presented in Figure 2 as violin plots with individual points. a-g Average transmembrane domain length and number of palmitoyls per protein ($N_{\text{palmitoyl}}/\text{protein}$) for (a, b) single-pass and (c, d) multi-pass transmembrane proteins, and (e, f, g) number of myristoyl, palmitoyl, and prenyl groups on peripheral membrane proteins were compared. Error bars represent SEM, number of proteins in each category can be found in Supplementary Table 1. A one-way ANOVA was performed to compare structural features between each data set, and comparisons were evaluated using the Sidak multiple comparisons correction (*, $p < 0.05$, **, $p < 0.01$, ***, $p < 0.001$).

2. The rationale behind the use of 4TMD and 12 TMD in Figure 4 is missing from the text.

We have added additional text to better explain the rationale behind the use of the 4TMD and 12TMD proteins.

We reasoned that these proteins are ideal candidates to test the core hypothesis in this study because they are completely synthetic, and therefore we expected that they would be trafficked and loaded into EVs based primarily on physical interactions with membranes, rather than through evolved protein-protein interactions and trafficking mechanisms. Furthermore, evaluating both the 4TMD and 12TMD proteins would enable us to test the extent to which the trends observed for the LAT proteins held for proteins with different numbers of transmembrane domains and different hydrophobic surface area.

3. Supplementary figures 10 and 12 may be redundant.

Thank you for identifying this issue. We mistakenly placed the wrong figure in Figure 12 (Now Fig. S17). This has been corrected.

References cited in this document:

1. Vial, C. & Evans, R. J. Disruption of lipid rafts inhibits P2X1 receptor-mediated currents and arterial vasoconstriction. *J Biol Chem* **280**, 30705–30711 (2005).
2. Ye, C. *et al.* Src family kinases engage differential pathways for encapsulation into extracellular vesicles. *Journal of Extracellular Biology* **2**, e96 (2023).
3. Shelby, S. A., Castello-Serrano, I., Wisser, K. C., Levental, I. & Veatch, S. L. Membrane phase separation drives organization at B cell receptor clusters. *bioRxiv* 2021.05.12.443834 (2021) doi:10.1101/2021.05.12.443834.
4. Jeppesen, D. K. *et al.* Reassessment of Exosome Composition. *Cell* **177**, 428 (2019).
5. Kowal, J. *et al.* Proteomic comparison defines novel markers to characterize heterogeneous populations of extracellular vesicle subtypes. *Proc Natl Acad Sci U S A* **113**, E968–E977 (2016).
6. Durcin, M. *et al.* Characterisation of adipocyte-derived extracellular vesicle subtypes identifies distinct protein and lipid signatures for large and small extracellular vesicles. *J Extracell Vesicles* **6**, (2017).
7. Mathieu, M. *et al.* Specificities of exosome versus small ectosome secretion revealed by live intracellular tracking of CD63 and CD9. *Nature Communications* 2021 12:1 **12**, 1–18 (2021).
8. Skotland, T., Sandvig, K. & Llorente, A. Lipids in exosomes: Current knowledge and the way forward. *Prog Lipid Res* **66**, 30–41 (2017).
9. Diaz-Rohrer, B. B., Levental, K. R., Simons, K. & Levental, I. Membrane raft association is a determinant of plasma membrane localization. *Proc Natl Acad Sci U S A* **111**, 8500–8505 (2014).
10. Sezgin, E. *et al.* Elucidating membrane structure and protein behavior using giant plasma membrane vesicles. *Nat Protoc* **7**, 1042–1051 (2012).
11. Lorent, J. H. *et al.* Structural determinants and functional consequences of protein affinity for membrane rafts. *Nature Communications* 2017 8:1 **8**, 1–10 (2017).

12. Duncan, J. A. & Gilman, A. G. A cytoplasmic acyl-protein thioesterase that removes palmitate from G protein alpha subunits and p21(RAS). *J Biol Chem* **273**, 15830–15837 (1998).
13. Kutzleb, C. *et al.* Paralemmin, a Prenyl-Palmitoyl–anchored Phosphoprotein Abundant in Neurons and Implicated in Plasma Membrane Dynamics and Cell Process Formation. *Journal of Cell Biology* **143**, 795–813 (1998).
14. Steinkühler, J. *et al.* Controlled division of cell-sized vesicles by low densities of membrane-bound proteins. *Nature Communications* **2020 11:1 11**, 1–11 (2020).
15. Yurtsever, D. & Lorent, J. H. Structural Modifications Controlling Membrane Raft Partitioning and Curvature in Human and Viral Proteins. *Journal of Physical Chemistry B* **124**, 7574–7585 (2020).
16. Levental, I., Lingwood, D., Grzybek, M., Coskun, Ü. & Simons, K. Palmitoylation regulates raft affinity for the majority of integral raft proteins. *Proc Natl Acad Sci U S A* **107**, 22050–22054 (2010).
17. Sezgin, E., Levental, I., Mayor, S. & Eggeling, C. The mystery of membrane organization: composition, regulation and roles of lipid rafts. *Nature Reviews Molecular Cell Biology* **2017 18:6 18**, 361–374 (2017).
18. Lorent, J. H. *et al.* Plasma membranes are asymmetric in lipid unsaturation, packing and protein shape. *Nat Chem Biol* **16**, 644–652 (2020).
19. Villamizar, O. *et al.* Mesenchymal Stem Cell exosome delivered Zinc Finger Protein activation of cystic fibrosis transmembrane conductance regulator. *J Extracell Vesicles* **10**, (2021).
20. Lain.ček, D., Lebar, T. & Jerala, R. Transcription activator-like effector-mediated regulation of gene expression based on the inducible packaging and delivery via designed extracellular vesicles. *Biochem Biophys Res Commun* **484**, 15–20 (2017).
21. Lain.ček, D. *et al.* Delivery of an Artificial Transcription Regulator dCas9-VPR by Extracellular Vesicles for Therapeutic Gene Activation. *ACS Synth Biol* **7**, 2715–2725 (2018).
22. Llorente, A. *et al.* Molecular lipidomics of exosomes released by PC-3 prostate cancer cells. *Biochimica et Biophysica Acta (BBA) - Molecular and Cell Biology of Lipids* **1831**, 1302–1309 (2013).
23. Haraszti, R. A. *et al.* High-resolution proteomic and lipidomic analysis of exosomes and microvesicles from different cell sources. *J Extracell Vesicles* **5**, (2016).

REVIEWER COMMENTS

Reviewer #1 (Remarks to the Author):

The authors addressed most of the comments but the following concerns still remain.

1. Deriving statistics (e.g., error bars, p values) from a sample size less than three ($n < 3$) is not considered appropriate, and such statistical measures must be excluded in these instances. Consequently, the conclusions drawn from most of the figures (e.g., Fig. 3f, 4d, 5d, e, f, g) lack statistical support. Additionally, some error bars, notably in Fig. 5e, appear to be quite large. The authors are strongly encouraged to offer more substantial justifications, going beyond statements like 'carefully designed the experiments' or 'limited our conclusions to conservative statements,' in order to clarify the rationale behind justifying statistical conclusions based on a sample size of $n = 2$ independent experiments within the context of their studies. Otherwise, the results from $n = 3$ experiments should be provided to support statistical conclusions.

2. The authors endeavored to establish a mechanistic link between raft association and extracellular vesicle (EV) loading in cells through treatment with Filipin III. However, they observed that the drug induced toxicity in HEK cells at 1 μM and had no effect at 0.5 μM . Despite citing other studies to support their claims, the mechanistic link remains unclear within the context of their research. To establish this connection, exploring alternative approaches such as using different drugs or employing siRNA against lipid raft pathways could be considered.

Reviewer #2 (Remarks to the Author):

The authors have addressed most of my concerns and I would suggest publication after the authors confirm one of their statements by a small experiment or rephrase their statement.

Regarding question 6 (reviewer 2):

The authors argue that the PPF construct (containing 2 palmitoyls) and the PM construct (containing 1 palmitoyl) would be associated differently to the membrane because "palmitoylation is reversible." Because it is reversible, both palmitoyl groups could be reversed for the PPF construct which would simply leave one farnesyl group. The PM construct would have one myristoyl group left if the palmitoyl group is detached.

Right now the statement reads "We hypothesize that the PPF-tagged synTFs may be more tightly bound to membranes than PMtagged

synTFs in reporter cells because the PPF tag is composed of 3 lipids (2 palmitic acids and 1 farnesyl) compared to two for PM (1 myristic and 1 palmitic acid)¹³."

If the authors want to prove this statement why not prove this point with a fluorescent PPF/PM construct?

Otherwise, if they do not want to prove this point, it would be better to say that they cannot explain the difference right now.

Reviewer #3 (Remarks to the Author):

The authors have answered all concerns raised by this reviewer.

REVIEWER COMMENTS

Reviewer #1 (Remarks to the Author):

The authors addressed most of the comments but the following concerns still remain.

1. Deriving statistics (e.g., error bars, p values) from a sample size less than three ($n < 3$) is not considered appropriate, and such statistical measures must be excluded in these instances. Consequently, the conclusions drawn from most of the figures (e.g., Fig. 3f, 4d, 5d, e, f, g) lack statistical support. Additionally, some error bars, notably in Fig. 5e, appear to be quite large. The authors are strongly encouraged to offer more substantial justifications, going beyond statements like 'carefully designed the experiments' or 'limited our conclusions to conservative statements,' in order to clarify the rationale behind justifying statistical conclusions based on a sample size of $n = 2$ independent experiments within the context of their studies. Otherwise, the results from $n = 3$ experiments should be provided to support statistical conclusions.

We thank the reviewer for this comment; in response, we have carefully reviewed our statistical analyses and conclusions drawn from these analyses. To employ a statistical test, one must ensure that the data satisfies underlying assumptions of the chosen statistical test. Formally, all statistical tests employed in this study can be used when $n=2$; however upon checking that our data also satisfied the assumptions upon which such tests are based, we found some instances where particular assumptions were not met, and we have adjusted the analyses accordingly with no major impacts on the key conclusions of our study (outlined in detail below). Thus, we respectfully disagree that deriving statistical measures from $n < 3$ is always inappropriate, and we provide a thorough justification for this position here:

On performing statistical analyses when $n=2$:

All statistical tests performed in this study can be performed with $n=2$.

One can estimate the error of a distribution for $n=2$. The formula for standard error of the mean is as follows:

a

$$S. E. M. = \frac{\sigma}{\sqrt{N}}$$

where σ represents the standard deviation and n represents the sample size. When $n=2$, SEM simplifies to half of the range^{1,2}.

Further, the statistical tests that we have employed in this study, such as t tests and ANOVA, are still valid for small samples sizes as long as the assumptions required to perform the test (for any sample size)

are met. Indeed, these tests have been shown to be accurate for small samples sizes in typical

cases^{1,3}. Examination of assumptions required for statistical analysis:

We are grateful to the reviewer for prompting us to review our statistical analysis choices, in general, because this drove us to carefully reexamine all statistical tests performed in our study to ensure they were appropriate for the dataset being analyzed. Most of the analyses did not change as a result of this review. In some cases, assumptions for particular tests were not met, and in this revised manuscript, we employed different (and appropriate) statistical methodologies as a result. In the following table, we have listed all statistical tests performed in this study and documented the relevant assumptions upon which such tests

are based. The raw output from GraphPad, the software used to perform these analyses, has been incorporated into Source Data accompanying this manuscript, and the following table has been added to the manuscript as Supplementary Table 9.

Supplementary Table 9. Summary of statistical tests used in this study and their assumptions.

Figure	Statistical Test	Assumptions		
		n are independent?	Homogeneity of variances met?	Normally distributed residuals met?
1c	1-way ANOVA + Sidak	Yes	Yes, Brown-Forsythe	Yes, D'Agostino-Pearson
1d	Unpaired t test	Yes	Yes, F test	Yes, Shapiro-Wilk
2a	Kruskal-Wallis	Yes	Not required for test	Not required for test
2b	Kruskal-Wallis	Yes	Not required for test	Not required for test
2c	Kruskal-Wallis	Yes	Not required for test	Not required for test
2d	Kruskal-Wallis	Yes	Not required for test	Not required for test
2e	Kruskal-Wallis	Yes	Not required for test	Not required for test
2f	Kruskal-Wallis	Yes	Not required for test	Not required for test
2g	Kruskal-Wallis	Yes	Not required for test	Not required for test
S5a	Kruskal-Wallis	Yes	Not required for test	Not required for test
S5b	Kruskal-Wallis	Yes	Not required for test	Not required for test
S5c	Kruskal-Wallis	Yes	Not required for test	Not required for test
S5d	Kruskal-Wallis	Yes	Not required for test	Not required for test
S5e	Kruskal-Wallis	Yes	Not required for test	Not required for test
S5f	Kruskal-Wallis	Yes	Not required for test	Not required for test
S5h	Kruskal-Wallis	Yes	Not required for test	Not required for test
S6a	Kruskal-Wallis	Yes	Not required for test	Not required for test
S6b	Kruskal-Wallis	Yes	Not required for test	Not required for test
S6c	Kruskal-Wallis	Yes	Not required for test	Not required for test
S6d	Kruskal-Wallis	Yes	Not required for test	Not required for test
S6e	Kruskal-Wallis	Yes	Not required for test	Not required for test
S6f	Kruskal-Wallis	Yes	Not required for test	Not required for test
S6g	Kruskal-Wallis	Yes	Not required for test	Not required for test
S7a	Kruskal-Wallis	Yes	Not required for test	Not required for test
S7b	Kruskal-Wallis	Yes	Not required for test	Not required for test
S7c	Kruskal-Wallis	Yes	Not required for test	Not required for test
S7d	Kruskal-Wallis	Yes	Not required for test	Not required for test
S7e	Kruskal-Wallis	Yes	Not required for test	Not required for test
S7f	Kruskal-Wallis	Yes	Not required for test	Not required for test
S7g	Kruskal-Wallis	Yes	Not required for test	Not required for test
S8a	Kruskal-Wallis	Yes	Not required for test	Not required for test
S8b	Kruskal-Wallis	Yes	Not required for test	Not required for test
S8c	Kruskal-Wallis	Yes	Not required for test	Not required for test
S8d	Kruskal-Wallis	Yes	Not required for test	Not required for test
S8e	Kruskal-Wallis	Yes	Not required for test	Not required for test
S8f	Kruskal-Wallis	Yes	Not required for test	Not required for test
S8h	Kruskal-Wallis	Yes	Not required for test	Not required for test

S9a	Kruskal-Wallis	Yes	Not required for test	Not required for test
S9b	Kruskal-Wallis	Yes	Not required for test	Not required for test
S9c	Kruskal-Wallis	Yes	Not required for test	Not required for test
S9d	Kruskal-Wallis	Yes	Not required for test	Not required for test
S9e	Kruskal-Wallis	Yes	Not required for test	Not required for test
S9f	Kruskal-Wallis	Yes	Not required for test	Not required for test
S10a	Kruskal-Wallis	Yes	Not required for test	Not required for test
S10b	Kruskal-Wallis	Yes	Not required for test	Not required for test
S10c	Kruskal-Wallis	Yes	Not required for test	Not required for test
S10d	Kruskal-Wallis	Yes	Not required for test	Not required for test
S10e	Kruskal-Wallis	Yes	Not required for test	Not required for test
S10f	Kruskal-Wallis	Yes	Not required for test	Not required for test
S10g	Kruskal-Wallis	Yes	Not required for test	Not required for test
3c	1-way ANOVA + Tukey	Yes	Yes, Brown-Forsythe	Yes, D'Agostino-Pearson
3e	Welch's ANOVA + Dunnnett T3	Yes	Not required for test	Yes, D'Agostino-Pearson
3f	2-way ANOVA + Dunnnett	Yes	Yes, Spearman	Yes, D'Agostino-Pearson
3g	2-way ANOVA + Dunnnett	Yes	Yes, Spearman	Yes, D'Agostino-Pearson
S13a	Ordinary 2-way ANOVA	Yes	Yes, Spearman	Yes, Kolmogorov-Smirnov
4b,4TMD	1-way ANOVA + Tukey	Yes	Yes, Brown-Forsythe	Yes, D'Agostino-Pearson
4b,12TMD	Unpaired t test	Yes	Yes, F test	Yes, D'Agostino-Pearson
4c	1-way ANOVA + Tukey	Yes	Yes, Brown-Forsythe	Yes, Shapiro-Wilk
4d,4TMD	2-way ANOVA + Tukey	Yes	Yes, Spearman	Yes, D'Agostino-Pearson
4d,12TMD	2-way ANOVA + Tukey	Yes	Yes, Spearman	Yes, D'Agostino-Pearson
4e,4TMD	2-way ANOVA + Tukey	Yes	Yes, Spearman	Yes, D'Agostino-Pearson
4e,12TMD	2-way ANOVA + Tukey	Yes	Yes, Spearman	Yes, D'Agostino-Pearson
S13b	Ordinary 2-way ANOVA	Yes	Yes, Spearman	Yes, Kolmogorov-Smirnov
5b	Welch's ANOVA + Dunnnett T3	Yes	Not required for test	Yes, D'Agostino-Pearson
5c	1-way ANOVA + Tukey	Yes	Yes, Brown-Forsythe	Yes, D'Agostino-Pearson
5d	2-way ANOVA + Dunnnett	Yes	Yes, Spearman	Yes, Anderson-Darling
S13c	Ordinary 2-way ANOVA	Yes	Yes, Spearman	Yes, D'Agostino-Pearson
7b	1-way ANOVA of log transform + Dunnnett	Yes	Yes, Brown-Forsythe	Yes, D'Agostino-Pearson
7c	1-way ANOVA + Dunnnett	Yes	Yes, Brown-Forsythe	No per D'Agostino- Pearson, but justifying test use because residuals are expected to be Gaussian (see 5d)
7f	1-way ANOVA + Dunnnett	Yes	Yes, Brown-Forsythe	Yes, D'Agostino-Pearson

Examination of specific claims from select studies with n = 2:

With regards to the reviewer's assertion that “the conclusions drawn from most of the figures (e.g., Fig. 3f, 4d, 5d, e, f, g) lack statistical support,” we respectfully disagree.

We state the following with regards to Figure 3f:

“Notably, wild-type LAT loaded into vesicles from both the HS-EV fraction and UC-EV fraction to a greater extent than did most LAT mutants (Fig. 3f, Supplementary Fig. 15). Protein loading for the C26A and dCore mutants was less than observed for wild-type LAT, which is consistent with the plasma membrane localization and raft partitioning experiments (Fig. 3c, 3e, Supplementary Fig. 13).”

These statements are supported by statistical tests which are described above in “Examination of assumptions required for statistical analysis.”

We state the following regarding Figure 4d:

“Transmembrane domain length positively correlated with EV loading and EV partitioning for the proteins with four transmembrane domains (Fig. 4d, 4e, Supplementary Fig. 16). However, this trend was not observed for the hexameric transmembrane hairpin designs (12TMD). This outcome that transmembrane domain length is not always predictive of EV loading is consistent with our bioinformatic data, since transmembrane domain length did not correlate with raft or EV association for multi-pass transmembrane proteins (Fig. 2c).”

These statements are supported by statistical tests which are described above in “Examination of assumptions required for statistical analysis.”

We state the following regarding Figure 5d:

“Lipidation increased HaloTag loading relative to soluble HaloTag for all conditions in both EV populations, achieving up to 400 proteins per vesicle and a loading improvement of up to ~4x over passive loading (Fig. 5d, Supplementary Fig. 17).”

These statements are supported by statistical tests and are described above in “Examination of assumptions required for statistical analysis.” Figures 5e, f, and g were all found to fail homogeneity of variance tests, and we have therefore removed indications of statistical significance related to those figure panels and have reviewed the associated conclusions to reflect the removal of these tests.

In summary, we have carefully reevaluated our statistical methodologies and associated claims to support our interpretations of the data—not just for those in which $n = 2$, but for all experiments in this manuscript. We believe these steps have improved the quality and rigor of our science, and we are grateful for the reviewer's suggestion to explicitly provide justification for the statistical analyses used.

2. The authors endeavored to establish a mechanistic link between raft association and extracellular vesicle (EV) loading in cells through treatment with Filipin III. However, they observed that the drug induced toxicity in HEK cells at 1 μM and had no effect at 0.5 μM . Despite citing other studies to support their claims, the

mechanistic link remains unclear within the context of their research. To establish this connection, exploring alternative approaches such as using different drugs or employing siRNA against lipid raft pathways could be considered.

We agree with the reviewer that it would be valuable to demonstrate that by disrupting lipid rafts, protein loading into EVs is reduced. As the reviewer noted, we unfortunately found that Filipin III was too toxic in our system (i.e., dose required, time course, etc.) to disrupt rafts without compromising cell viability. The suggestion to use alternate drugs or siRNA against lipid raft pathways is intriguing, and one could imagine an impactful study that systematically determines the contributions of individual lipid raft pathways or lipidation machinery towards the loading of one or more lipidation tags. Such mechanistic studies represent an important future direction for such research, and we respectfully argue that such analysis is out of the scope of the present, foundational work.

We would highlight that through the careful selection of different transmembrane and peripheral proteins with differential lipid raft localization and cellular trafficking, we were still able to establish a connection between lipid-protein interactions and EV loading, particularly for certain protein classes (Figure 6, S18). As we discuss, cite, and contextualize in the discussion, our work is concordant with observations made by others that have used Filipin III to evaluate the loading of lipidated cargo into EVs in an alternate cell type.

Reviewer #2 (Remarks to the Author):

The authors have addressed most of my concerns and I would suggest publication after the authors confirm one of their statements by a small experiment or rephrase their statement.

Thank you for your review and thoughtful comments. They have greatly improved the quality of this manuscript!

Regarding question 6 (reviewer 2):

The authors argue that the PPF construct (containing 2 palmitoyls) and the PM construct (containing 1 palmitoyl) would be associated differently to the membrane because "palmitoylation is reversible." Because it is reversible, both palmitoyl groups could be reversed for the PPF construct which would simply leave one farnesyl group. The PM construct would have one myristoyl group left if the palmitoyl group is detached.

Right now the statement reads "We hypothesize that the PPF-tagged synTFs may be more tightly bound to membranes than PM-tagged synTFs in reporter cells because the PPF tag is composed of 3 lipids (2 palmitic acids and 1 farnesyl) compared to two for PM (1 myristic and 1 palmitic acid)¹³."

If the authors want to prove this statement why not prove this point with a fluorescent PPF/PM construct?

Otherwise, if they do not want to prove this point, it would be better to say that they cannot explain the difference right now.

The proposed experiment is an excellent suggestion and one we are very interested in evaluating in the future. As suggested, we have modified the statement to clarify that the reason for the difference in EV loading and EV bioactivity between the PM and PPF constructs is unknown, but we retained some speculation as to potential reasons for the difference in function which warrant exploration in future studies.

Interestingly, the PPF tag conferred the highest loading of protein cargo, and it performed better than the PM tag when directly expressed in reporter cells but did not enable functional delivery of our synTF. Presently, we cannot explain the discrepancy between loading and EV-mediated activity (i.e., reporter gene expression) between the PM and PPF tags and believe this is an interesting area for future research. The difference in functional delivery may possibly arise from the ability of each tag to enable transport to the nucleus. One potential hypothesis is that the PPF-tagged synTFs may be more tightly bound to membranes than PM-tagged synTFs in reporter cells because the PPF tag is composed of 3 lipids (2 palmitic acids and 1 farnesyl) compared to two for PM (1 myristic and 1 palmitic acid)⁵⁸. Alternatively, differences in delivery could be explained by inherent trafficking differences between lipidation tags—indeed, the SH4 domain of Lyn, the protein domain from which the PM tag is derived, drives association of Lyn with the plasma membrane and internal membranes such as the Golgi and nuclear membranes^{59–61}.

Reviewer #3 (Remarks to the Author):

The authors have answered all concerns raised by this reviewer.

Thank you for your time and thoughtful review of our work.

Citations in this response document

1. Statistics with n=2 - FAQ 591 - GraphPad. <https://www.graphpad.com/support/faqid/591/>.
2. Cumming, G., Fidler, F. & Vaux, D. L. Error bars in experimental biology. *J Cell Biol* **177**, 7 (2007).
3. Winter, J. C. F. de. Using the Student's t-test with extremely small sample sizes. *Practical Assessment, Research, and Evaluation* **18**, (2013).

REVIEWERS' COMMENTS

Reviewer #1 (Remarks to the Author):

The authors have made significant efforts to extensively justify the use of $n = 2$ in several figures, supported by Supplementary Table 9. However, I leave the final decision regarding adherence to Nature's editorial policies in deriving statistics from $n < 3$ to the editorial team.

Reviewer #2 (Remarks to the Author):

The authors have adressed all my concerns and I recommend hence publication.

Reviewer #1 (Remarks to the Author):

The authors have made significant efforts to extensively justify the use of $n = 2$ in several figures, supported by Supplementary Table 9. However, I leave the final decision regarding adherence to Nature's editorial policies in deriving statistics from $n < 3$ to the editorial team.

Reviewer #2 (Remarks to the Author):

The authors have addressed all my concerns and I recommend hence publication.

We thank both reviewers for their time and thoughtful comments and suggestions and have prepared the final revision documents.